# DART: Distribution-Aware Adaptive Relational Transfer for Adversarial Attacks against Closed-Source MLLMs

**Kaidi Hu**[1]  **Guancheng Wan**[2]  **Xiao Luo**[3]  **Ruigang Yang**[1]

## Abstract

This paper studies the critical problem of targeted adversarial attacks against closed-source MLLMs, which aim to generate highly transferable adversarial samples with open-source MLLMs. Previous approaches typically focus on maximizing the similarity of latent representations between adversarial samples and target samples. However, these approaches could overfit specific target samples with severely limited generalization ability to closed-source MLLMs. Towards this end, we propose a novel approach named Distribution-aware Adaptive Relational Transfer (DART) for adversarial attacks against closed-source MLLMs. The core of our DART is to adopt a statistical lens to characterize the intrinsic semantics of images for more generalized and robust alignment. In particular, each augmented image is considered an example from the intrinsic distribution of the original image. Then, we utilize non-parametric Energy Distance to measure the distribution divergence, which is naturally adopted for the semantic alignment in the hidden space. To further enhance transferability to specific target models, we learn a graph neural network (GNN) to explore the complex relations between source and target MLLMs on transferability and adaptively select surrogate models to maximize transferability across diverse targets. Extensive experiments on benchmark datasets validate the superior robustness and effectiveness of the proposed DART in comparison to various competing baselines.

[1]Global Institute of Future Technology, Shanghai Jiao Tong University, Shanghai, China [2]Wuhan University, Wuhan, China [3]Department of Statistics, University of Wisconsin-Madison, Madison, USA. Correspondence to: Ruigang Yang <ryang2@sjtu.edu.cn>, Xiao Luo <xiao.luo@wisc.edu>.

*Proceedings of the $43^{rd}$ International Conference on Machine Learning*, Seoul, South Korea. PMLR 306, 2026. Copyright 2026 by the author(s).

## 1. Introduction

Multimodal large language models (MLLMs) have achieved extensive breakthroughs recently. Models including GPT-4o and Claude-3.7 have shown outstanding performance in a range of vision-language tasks, such as visual reasoning (Li et al., 2024b; Park et al., 2025; Huang et al., 2025), image captioning (Salaberria et al., 2023; Li et al., 2024a; Sarto et al., 2025), and visual question answering (Luu et al., 2024; Özdemir & Akagündüz, 2024; Kuang et al., 2025). These models, especially widely adopted commercial closed-source versions, are increasingly entering high-risk domains such as autonomous driving (Cui et al., 2024; Fu et al., 2024a; Fime et al., 2025; Zeng et al., 2025), medical diagnosis (Jeong et al., 2024; Wang et al., 2025b), and content moderation (Ye et al., 2025; Fang et al., 2025). However, their widespread deployment introduces serious security challenges, with adversarial attacks posing a major threat that undermines the reliability and safety of these systems (Jiang et al., 2025).

Despite their strong capabilities, MLLMs often inherit the adversarial vulnerabilities of their visual encoders, making them susceptible to adversarial attacks (Gao et al., 2024; Liu et al., 2024). Among various attack scenarios, targeted transfer attacks on closed-source models are the most challenging and realistic. Unlike untargeted attacks that simply induce incorrect predictions, we focus on targeted attacks in which adversaries aim to manipulate MLLMs to produce specific, attacker-desired semantic captions. In this black-box setting, attackers have no access to the target model's internal information, such as the architecture or parameters. Instead, they craft adversarial examples on an open-source surrogate model, aiming for successful transfer to the target model (Byun et al., 2022).

Currently, transfer attack methods for MLLMs have made notable progress. The core approach typically generates perturbations by maximizing the similarity between adversarial and target samples in the latent space, leveraging open-source surrogate models such as CLIP and its variants. Several advanced approaches have recently emerged in this field. For example, M-Attack found that traditional uniformly distributed perturbations lack semantic information, often causing attack failures (Li et al., 2025). It introduced

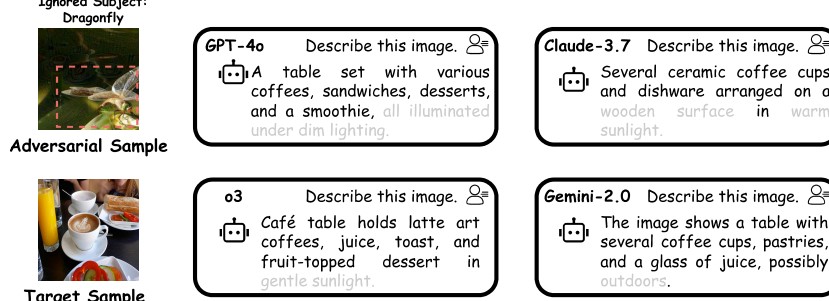

Figure 1. Examples of responses from closed-source MLLMs to targeted adversarial samples generated by DART. Bold text marks keywords that are semantically consistent with the target sample, whereas gray text highlights topics that the model overlooks.

an innovative yet simple random clipping strategy to embed local semantic details within perturbations, significantly improving attack effectiveness. FOA-Attack adopts a more sophisticated alignment strategy, aligning both global features and fine-grained local blocks using clustering and optimal transport techniques (Jia et al., 2025). It further incorporates a dynamic model-weighting strategy driven by loss convergence speed.

However, despite continuous technical advancements, these methods share a fundamental limitation: they primarily focus on point-wise feature alignment by minimizing the distance between individual adversarial and target feature vectors. This approach causes adversarial perturbations to overfit the surrogate model's specific feature space, a challenge conceptually illustrated in the left panel of Figure 2. This overfitting results in two key issues: **1) Poor Generalization:** The generated perturbations become overly dependent on the surrogate model's latent space, leading to semantic fragility and poor generalization. Consequently, these perturbations are difficult to transfer to closed-source target models with different architectures and parameters. **2) Lack of Adaptivity:** Existing ensemble strategies—such as the static uniform weighting in M-Attack or the heuristic dynamic weighting in FOA-Attack—fail to capture the complex, asymmetric, and task-dependent transfer relationships among different MLLMs. As a result, these methods cannot be considered truly adaptive.

To overcome these challenges, we introduce a novel attack framework, Distribution-aware Adaptive Relational Transfer (DART), designed to fundamentally redefine the paradigm of transfer attacks. The core concept of DART is to decompose the adversarial attack problem into two subproblems—generalizability and specificity—offering independent and principled solutions for each:

**(1) Transitioning from point-wise to distributional alignment to improve generalization.** Rather than matching individual feature vectors, we align the entire latent space distribution that captures an image's intrinsic semantic information. We hypothesize that an image's core semantic

information can be represented as a probability distribution, sampled via data augmentation techniques such as random cropping and scaling. Second, we employ the robust non-parametric statistical metric, Energy Distance, as a loss function to minimize distributional discrepancies between adversarial and target samples in the latent space. By aligning statistical distributions instead of individual feature points, DART produces perturbations that capture more fundamental and generalizable semantic information, effectively mitigating feature overfitting in surrogate models.

**(2) Moving from heuristic ensemble methods to relational ensembling for improved specialization.** To overcome the limitations of existing ensemble strategies, DART introduces an innovative GNN Attack Router. This component explicitly models the transferability of surrogate models across attack tasks using a heterogeneous graph. Through offline training, the GNN Attack Router learns complex transfer relationships among models and adaptively selects the optimal surrogate model ensemble for specific attack tasks. This shifts attack strategies from static or heuristic integration to a data-driven, highly specialized selection process tailored to specific objectives.

Figure 1 shows examples demonstrating the effectiveness of DART's adversarial attacks on closed-source MLLMs. The main contributions of this paper are as follows:

❶ *Paradigm Shift: From Point-Wise to Distributional Alignment.* We observe that overfitting in existing transfer attacks stems from point-wise feature matching. We propose a statistical perspective that models image semantics as a latent distribution and leverages manifold alignment to fundamentally improve generalization.

❷ *The DART Framework.* We introduce a unified optimization framework that integrates a non-parametric Energy-Distance objective for robust intrinsic feature mining and a GNN Attack Router for adaptive surrogate selection. This design simultaneously addresses challenges related to both generalization and specialization.

❸ *SOTA Performance and Semantic Robustness.* Extensive experimental evaluations demonstrate that DART

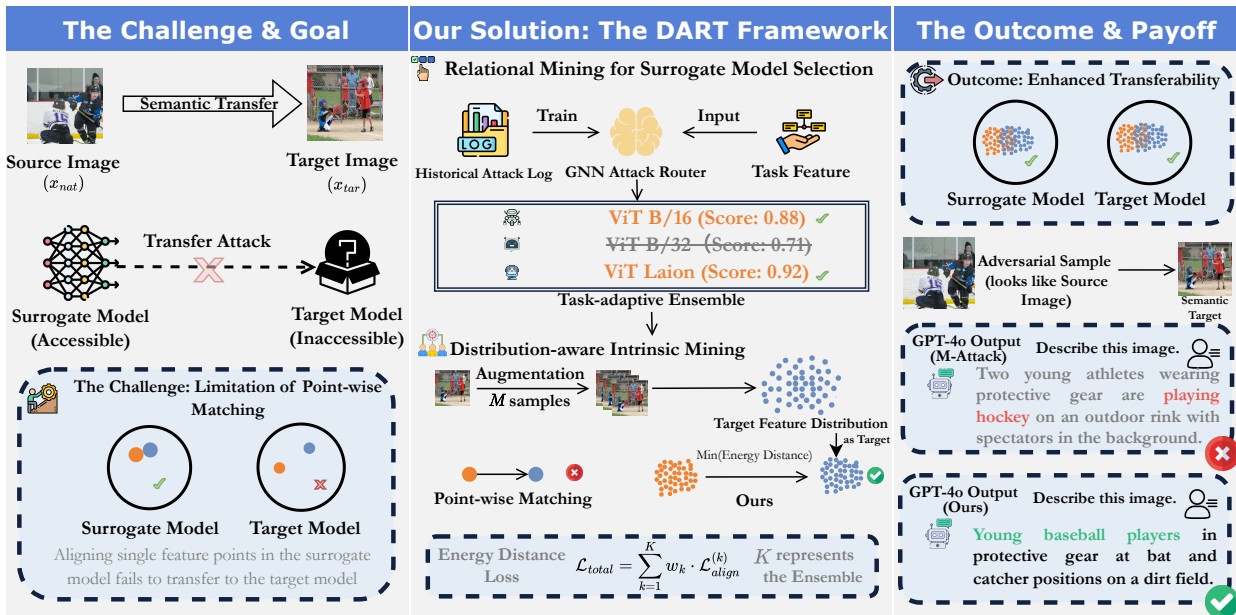

*Figure 2.* **Overview of the DART Framework.** It addresses the overfitting issue of point-wise matching (Left) by introducing a GNN Attack Router for adaptive ensemble and a Distribution-aware Intrinsic Mining module (Middle). This dual strategy successfully aligns adversarial samples with the target semantic distribution (Right).

establishes a new technical baseline. Compared with leading competing approaches, it achieves significant improvements in attack success rate on commercial black-box models, validating its practical effectiveness.

## 2. Related Work

**Transfer-based Adversarial Attacks on MLLMs.** Early studies showed that aligning image features in the latent space of surrogate models is an effective transfer attack method (Zhao et al., 2023). Later research mainly improved transferability through two strategies. One direction uses ensemble approaches, evolving from simple loss averaging to generating more robust perturbations by exploiting cross-model shared vulnerabilities (Dong et al., 2023). The other direction leverages semantic information to improve the alignment process. For example, M-Attack adds local semantic embeddings into perturbations through random cropping (Li et al., 2025), while FOA-Attack applies clustering and optimal transport for finer-grained feature matching, enhanced by dynamic loss weighting (Jia et al., 2025). Despite these advances, current approaches are still limited by their reliance on point-wise feature alignment (which risks overfitting to the surrogate model) and by heuristic or static integration rules. Our method addresses these limitations by adopting distribution alignment for better generalization and introducing a learning-based adaptive model selection strategy.

**Statistical Distances for Distribution Matching.** Moving from "point-to-point" to "distribution-to-distribution" alignment requires a robust metric to measure differences between high-dimensional empirical distributions. Metrics such as Maximum Mean Discrepancy (MMD) are common, but their performance is highly sensitive to kernel selection (Gretton et al., 2006; 2012). The Wasserstein Distance provides rich geometric information but is computationally intensive in high-dimensional MLLM latent spaces (Villani et al., 2008; Arjovsky et al., 2017). In contrast, we introduce the Energy Distance—a non-parametric metric that is theoretically sound, requires no hyperparameter tuning, and is computationally efficient (Rizzo & Székely, 2016; Zhang et al., 2024a), making it particularly well-suited for iterative high-dimensional optimization.

**Graph Neural Networks for Relational Modeling.** To overcome the limits of static or heuristic integration, we formulate surrogate model selection as a relational learning problem. We propose a GNN Attack Router to model complex, task-relevant transferable relationships among MLLMs. This approach is inspired by the success of GNNs in capturing high-order relational patterns across domains (Yu et al., 2022; Feng et al., 2025; Chen et al., 2025). By learning transfer patterns from historical data, our method enables a data-driven, adaptive surrogate model selection framework that outperforms fixed strategies in prior work, thereby improving attack specialization.

**Comparison with Traditional Attack Methods.** Traditional attack methods (e.g., FGSM and PGD) primarily exploit high-frequency pixel noise to cross static decision boundaries in closed-set classification, resulting in simple la-

bel misclassifications (Cao et al., 2025). In contrast, MLLM attacks operate in an open semantic space dominated by cross-modal alignment, where the attack objective shifts from simple misclassification to more complex forms of semantic hijacking or jailbreaking (Cui et al., 2025). Because the visual–language projector in MLLMs acts as a semantic filter that blocks simple pixel-level gradient noise, point-to-point feature-matching strategies developed for CNNs do not transfer directly (Wang et al., 2025a). Consequently, effective MLLM attacks must move beyond pixel-level perturbations and instead exploit distribution-level semantic alignment to breach the security barriers of multimodal interactions—a defining characteristic of this new generation of threat models (Rahmatullaev et al., 2025).

## 3. The Proposed DART

We target the black-box transfer attack setting where the adversary aims to generate an imperceptible perturbation $\delta$ ($||\delta||_p \leq \epsilon$) for a natural image $x_{nat}$, such that the target MLLM $M_{tgt}$ produces a description semantically equivalent to a target image $x_{tar}$. Since $M_{tgt}$ is inaccessible, we optimize $\delta$ on a set of open-source surrogate models $\mathcal{F} = \{f_{\theta_i}\}_{i=1}^T$.

### 3.1. Framework Overview

As shown in Figure 2, the DART framework addresses both generalization and specialization through a two-stage design. **1) Offline Relationship Mining:** We construct a heterogeneous graph using historical attack logs, including source-target pairs and surrogate outcomes, and pre-train a GNN-based Attack Router. The router is trained to predict the transferability of different surrogate combinations. **2) Online Attack Generation:** Given a new target pair $(x_{nat}, x_{tar})$, the router selects the Top-$K$ optimal surrogate combinations $\mathcal{F}^*$. We then optimize the perturbations by minimizing the Energy Distance between the latent distributions of adversarial and target samples. The latent distributions are estimated through data augmentation.

### 3.2. Distribution-Aware Intrinsic Mining for Representation Alignment

To overcome the limited generalization of prior work (Li et al., 2025; Jia et al., 2025), caused by overfitting to specific feature points in surrogate models, we propose a novel Distribution-aware Intrinsic Mining approach. Our method shifts the attack objective from reconstructing individual feature vectors to aligning latent distributions that capture an image's core semantics. We hypothesize that an image's intrinsic information can be represented as a probability distribution, which we approximate through empirical sampling with data augmentation. We employ the nonparametric Energy Distance (Rizzo & Székely, 2016; Zhang

et al., 2024a), a robust statistical metric, to minimize the divergence between the adversarial sample and the target sample's underlying distribution. By operating at the distributional rather than the feature-point level (as illustrated in the middle panel of Figure 2), our method produces adversarial perturbations less sensitive to the surrogate model's feature space, thereby substantially improving transferability and generalization, achieving successful alignment.

Our approach defines the semantic identity of an image as a latent distribution $P_\theta(x)$, rather than as a single latent vector. To approximate this distribution empirically, we apply $M$ random data augmentations $\mathcal{T}$ to the image $x$, which generates a batch of augmented views $\{x'_i = \mathcal{T}_i(x)\}_{i=1}^M$. Following standard practice in self-supervised learning, we use semantic-preserving transformations that alter low-level visual statistics while preserving high-level semantics, instantiated by RandomResizedCrop, RandomHorizontalFlip, and ColorJitter. This design principle aligns with our assumption that the augmented views are samples from the same intrinsic semantic distribution. For each view, we use an encoder $f_\theta$ to extract two representations: a global feature vector $g_i = G_\theta(x'_i)$ capturing high-level semantics, and local block features $L_i = L_\theta(x'_i)$ preserving fine details. Consequently, we obtain two empirical distributions for the target image $x_{tar}$: the global feature distribution $\mathcal{D}_G(x_{tar})$ and the local feature distribution $\mathcal{D}_L(x_{tar})$, defined as $\mathcal{D}_G(x) = \{G_\theta(x'_i)\}_{i=1}^M$ and $\mathcal{D}_L(x) = \{\mathcal{C}(L_\theta(x'_i))\}_{i=1}^M$, respectively, where $\mathcal{C}$ denotes $K$-means clustering with 10 centroids that aggregates local features. We adopt $K$-means for its simplicity, guaranteed convergence, and efficiency in iterative training, with 10 clusters chosen to balance the granularity of local pattern representation against computational overhead.

After defining the latent distribution, we generate adversarial perturbations by minimizing the Squared Cosine Energy Distance between the adversarial sample $x_{adv}$ and the target $x_{tar}$. The Squared Cosine Energy Distance is given by:

$$\mathcal{D}_c^2(F, G) = 2\mathbb{E}[d_c(\tilde{X}, \tilde{Y})] - \mathbb{E}[d_c(\tilde{X}, \tilde{X}')] - \mathbb{E}[d_c(\tilde{Y}, \tilde{Y}')], \tag{1}$$

where $\tilde{X}$, $\tilde{X}'$ and $\tilde{Y}$, $\tilde{Y}'$ denote independent and identically distributed unit vectors sampled from distributions $F$ and $G$, respectively. Intuitively, the Energy Distance captures the statistical difference between two distributions, $F$ and $G$. The first term, $\mathbb{E}[d_c(\tilde{X}, \tilde{Y})]$, represents the expected distance between samples from different distributions, whereas the latter two terms, $\mathbb{E}[d_c(\tilde{X}, \tilde{X}')]$ and $\mathbb{E}[d_c(\tilde{Y}, \tilde{Y}')]$, represent the expected distance within each distribution. Thus, minimizing $\mathcal{D}_c^2(F, G)$ reduces both cross-distribution and intra-distribution distances, achieving distribution-aware semantic alignment under finite-sample approximation rather than exact matching of the full distributions. With a moderate number of augmented views in a high-dimensional latent

space, this objective approximates rather than perfectly recovers the underlying distribution, while still providing a robust, non-parametric objective for aligning adversarial images with the semantic manifold of the target image.

The alignment loss $\mathcal{L}_{align}$ is composed of a global loss $\mathcal{L}_{global}$ and a local loss $\mathcal{L}_{local}$. For global features, we use a point-to-distribution Energy Distance. For local features, we calculate the average distribution-to-distribution Energy Distance. Specifically, we define the global alignment loss as $\mathcal{L}_{global} = E_{p \to d}(g_{adv}, \mathcal{D}_G(x_{tar}))$ and the local alignment loss as $\mathcal{L}_{local} = \frac{1}{M} \sum_{i=1}^{M} \mathcal{D}_c^2(\mathcal{C}(L_{adv}), C_i)$. We deliberately adopted two distinct alignment strategies. To capture global features ($\mathcal{L}_{global}$) representing overall scene semantics, we use point-to-distribution ($p \to d$) Energy Distance. This choice relies on the assumption that the high-level semantic vectors of adversarial images align with the central tendency of the target semantic distribution. For local features ($\mathcal{L}_{local}$) that capture fine-grained details and textures, a stricter distribution-to-distribution ($d \to d$) alignment is required. This ensures that the rich multimodal local patterns in adversarial samples match the target distribution, thereby preventing semantic mismatches at fine-grained levels. This dual strategy ensures both high-level conceptual consistency and fine-grained texture fidelity.

**Remark on the role of intra-distribution terms.** A natural question is whether maximizing the intra-distribution term on the adversarial side, $\mathbb{E}[d_c(\tilde{X}, \tilde{X}')]$, in Equation 1 drives adversarial features apart and consequently leads to semantic divergence. In our framework, this concern does not arise. For the global branch ($\mathcal{L}_{global}$), adversarial samples reduce to single points (i.e., a Dirac measure). Consequently, $\mathbb{E}[d_c(\tilde{X}, \tilde{X}')] \equiv 0$, and the optimization process never maximizes the intra-distribution distance. For the local branch ($\mathcal{L}_{local}$), the intra-distribution terms play an important role by acting as a calibration mechanism that prevents local features from collapsing into a single point, analogous to the variance term in strictly proper scoring rules. Empirically, as shown in Appendix A, the intra-distribution value $E_{intra}(L_{adv})$ on the adversarial side remains stable throughout the optimization process and converges to the intrinsic variance of the target. It neither diverges nor collapses, which is fully consistent with the theoretical interpretation described above.

For an ensemble model with $K$ components, the total alignment loss $\mathcal{L}_{total}$ is defined as a dynamically weighted sum of the individual losses, where $\mathcal{L}_{align}^{(k)} = \mathcal{L}_{global}^{(k)} + \eta_k \mathcal{L}_{local}^{(k)}$.

$$\mathcal{L}_{total} = \sum_{k=1}^{K} w_k \cdot \mathcal{L}_{align}^{(k)},$$

$$w_k = K \cdot \frac{\exp(r_k/T)}{\sum_{j=1}^{K} \exp(r_j/T)}, \quad r_k = \frac{\mathcal{L}_{align}^{(k)}(t)}{\mathcal{L}_{align}^{(k)}(t-1)}. \tag{2}$$

This dynamic weighting mechanism is inspired by FOA-Attack (Jia et al., 2025), which plays a key role in generating transferable adversarial examples. The core idea is that, at each iteration, different surrogate models may present varying levels of attack difficulty. Models with slower loss convergence (i.e., higher $r_k$) are considered more "challenging" because their feature space is not yet well aligned with the target feature space. We apply the Softmax function, with temperature $T$ controlling the sharpness of the weight distribution, to the loss ratios $r_k$, thereby assigning greater weights $w_k$ to the more challenging models. The scaling factor $K$ ensures the weights fluctuate around 1.0. This strategy prevents optimization from converging prematurely to solutions that satisfy only the "simpler" models in the ensemble. Instead, it compels adversarial perturbations to focus on the most difficult-to-align feature dimensions. Consequently, the resulting perturbations avoid overfitting to any single model's characteristics and instead capture more robust, shared vulnerabilities. This greatly improves their transferability to unseen black-box target models $M_{tgt}$. In our experiments, we set the temperature $T=1$.

Intuitively, distribution alignment reduces gradient variance by averaging over augmented views rather than relying on a single target point. We formalize this argument in Appendix A using the statistical properties of Energy Distance. In practice, we adopt a cosine-based distance for stable, scale-invariant optimization in high-dimensional latent spaces. Empirical validation is reported in Section 4.3.

### 3.3. Relational Mining for Surrogate Model Selection

Traditional ensemble attacks usually rely on static strategies (e.g., uniform weighting) or heuristic strategies (e.g., loss-based) to combine surrogate models. However, these strategies cannot capture the complex, asymmetric, and task-dependent transferability relationships among different MLLMs. To address the limitations of static or heuristic ensemble strategies, we propose a GNN Attack Router that adaptively selects surrogate models through learning. This design transforms model selection from a manual, rule-based task into a data-driven relational learning problem. We represent the complex, task-dependent transferability between surrogate models and attack tasks using a heterogeneous graph. By pre-training a Graph Neural Network (GNN) on historical attack data, the GNN Attack Router learns to predict the most effective combinations of surrogate models for new, unseen attack tasks. In practice, the router concentrates computation on surrogate models with the highest predicted success rates, which improves transferability to the target MLLMs.

The core of this mechanism is a heterogeneous graph $\mathcal{G} = (\mathcal{V}, \mathcal{E})$, built from historical attack logs. The graph contains two types of nodes: (i) Task nodes ($v_k \in \mathcal{V}_{task}$),

with features formed by concatenating source and target image embeddings; and (ii) Model nodes ($v_m \in \mathcal{V}_{model}$), represented by one-hot encodings. If model $m$ is used to perform task $k$, an edge ($v_k, v_m$) is created with weight $w_{km}$, which represents the empirically measured transfer success score (computed with GPTScore (Fu et al., 2024b)). Crucially, these transfer scores are derived from historical attack evaluations where Qwen2-VL-7B serves as the log-collection target MLLM, which is intentionally disjoint from the target MLLMs evaluated in Section 4.2. During testing, we directly deploy the trained router to guide attacks against other disjoint target MLLMs, without requiring any target-specific queries or retraining. This approach reflects a practical zero-shot deployment scenario, in which the router generalizes from historical transfer behavior to unseen target architectures and interfaces, thereby mitigating the cold-start problem. This graph captures the complex dependencies between attack targets and model capabilities.

The GNN Attack Router conducts offline training on the graph to predict transfer scores. The architecture consists of an embedding layer, a heterogeneous encoder for message passing, and an MLP-based edge predictor. The training objective is to minimize the mean squared error between the predicted score $\hat{w}_{km}$ and the true score $w_{km}$. The loss function is defined as: $\mathcal{L} = \frac{1}{|\mathcal{E}_{train}|} \sum_{(v_k, v_m) \in \mathcal{E}_{train}} (\hat{w}_{km} - w_{km})^2$. Once training is complete, the GNN Attack Router can be deployed for online inference. For a new attack task, we query the GNN Attack Router to predict transfer scores for all available surrogate models and select the Top-$K$ models with the highest scores to form a specialized ensemble, $\mathcal{F}^*$. Our experiments in Section 4.3 show that $K=2$ achieves optimal performance. This data-driven, adaptive selection process ensures that our attacks both generalize well and are precisely tailored to the specific task.

## 4. Experiments

### 4.1. Datasets and Implementation Details

We build on prior research (Jia et al., 2025; Li et al., 2025) by using 1,000 source images from the NIPS 2017 competition (Kurakin et al., 2018) and 1,000 target images from the MS COCO validation set (Lin et al., 2014). We train the GNN Attack Router on the first 800 image pairs and use the remaining 200 pairs for evaluation. The surrogate model pool consists of three representative CLIP variants (Ilharco et al., 2021) (detailed in Appendix D). For evaluation, we follow Li et al. (2025) and employ LLM-as-a-judge (Zheng et al., 2023), where the same MLLM generates captions for both adversarial samples and target images. We then use the GPTScore method (Fu et al., 2024b) to calculate similarity scores between the two sets of captions. We report Attack Success Rate (ASR), Average Similarity Score (AvgSim),

and Keyword Matching Rate (KMR). *Crucially, unlike traditional latent-distortion metrics, ours directly evaluate the semantic consistency of generated captions to reflect downstream performance.* For complete reproducibility, detailed metric definitions, attack hyperparameters, and surrogate architectures are provided in Appendix D.

### 4.2. Comparison with State-of-the-Art Methods

To demonstrate DART's superiority, we conducted extensive comparisons with several recent state-of-the-art attack methods: AttackVLM (Zhao et al., 2023), AdvDiffVLM (Guo et al., 2024), SSA-CWA (Dong et al., 2023), AnyAttack (Zhang et al., 2024b), M-Attack (Li et al., 2025), and FOA-Attack (Jia et al., 2025). We evaluated performance on both open-source and closed-source MLLMs.

**Attack Performance on Open-Source MLLMs.** As shown in Table 1, DART consistently and significantly outperforms all competing methods across six representative open-source MLLMs. For example, on LLaVA-1.5-7B, DART achieves an ASR of 91.5%, exceeding the previous method, FOA-Attack, by 2.0%. Notably, on more advanced models such as Qwen2.5-VL-7B, DART achieves an ASR of 86.5%, a substantial improvement over FOA-Attack's 80.0%. This result highlights the strong generalization ability of our distribution-level alignment approach, which mitigates overfitting to surrogate models and enables effective transfer across diverse open-source architectures. Moreover, the consistently high AvgSim scores indicate that adversarial samples generated by DART preserve strong semantic consistency with target images.

**Attack Performance on Closed-Source MLLMs.** The primary challenge of this work is attacking proprietary closed-source MLLMs. Table 2 reports the attack results on five state-of-the-art commercial models. DART shows strong effectiveness. On GPT-4o and GPT-4.1, DART achieved ASRs of 89.5% and 89.0%, respectively, establishing a new milestone. Notably, on models known for their robustness—Claude-3.5 and Claude-3.7—DART reached ASRs of 20.0% and 32.0%, respectively. These results represent a consistent improvement over FOA-Attack (17.0% and 23.5%) and M-Attack (19.5% and 25.0%). These findings strongly support our core hypotheses: 1) aligning with latent distributions, rather than individual feature points, yields more robust and transferable perturbations; and 2) the GNN Attack Router effectively identifies optimal surrogate model combinations for specific tasks, thereby enhancing attack specialization.

**Effectiveness of Attacks on MLLMs with Enhanced Reasoning Capabilities.** We further evaluated DART on MLLMs with enhanced reasoning capabilities, which are typically considered more robust. As shown in Table 3, DART continued to demonstrate strong performance. On o3,

*Table 1.* Performance comparison on representative open-source MLLMs. Green subscripts denote gains over FOA-Attack.

| Method | Model | LLaVA-1.5-7B | | LLaVA-1.6-7B | | Qwen2.5-VL-3B | | Qwen2.5-VL-7B | | Gemma-3-4B | | Gemma-3-12B | |
|---|---|---|---|---|---|---|---|---|---|---|---|---|---|
| | | ASR | AvgSim | ASR | AvgSim | ASR | AvgSim | ASR | AvgSim | ASR | AvgSim | ASR | AvgSim |
| AttackVLM | VIT-B/16 | 4.0 | 0.03 | 3.5 | 0.02 | 3.0 | 0.02 | 3.0 | 0.02 | 2.0 | 0.02 | 4.5 | 0.02 |
| | VIT-B/32 | 3.0 | 0.03 | 3.0 | 0.02 | 2.5 | 0.02 | 1.5 | 0.02 | 1.0 | 0.02 | 4.0 | 0.03 |
| | Laion | 3.0 | 0.03 | 4.5 | 0.03 | 2.5 | 0.02 | 3.0 | 0.03 | 1.5 | 0.02 | 4.5 | 0.03 |
| AdvDiffVLM | Ensemble | 3.5 | 0.03 | 4.0 | 0.03 | 5.0 | 0.03 | 4.0 | 0.04 | 3.5 | 0.04 | 5.5 | 0.03 |
| SSA-CWA | Ensemble | 3.0 | 0.02 | 4.0 | 0.04 | 2.5 | 0.03 | 2.0 | 0.03 | 2.0 | 0.02 | 2.5 | 0.03 |
| AnyAttack | Ensemble | 16.5 | 0.09 | 16.0 | 0.09 | 11.0 | 0.06 | 18.0 | 0.10 | 7.5 | 0.05 | 8.5 | 0.06 |
| M-Attack | Ensemble | 85.5 | 0.55 | 85.5 | 0.54 | 63.0 | 0.36 | 74.0 | 0.43 | 48.0 | 0.26 | 47.5 | 0.25 |
| FOA-Attack | Ensemble | 89.5 | 0.56 | 91.5 | 0.57 | 66.0 | 0.41 | 80.0 | 0.48 | 50.5 | 0.28 | 51.5 | 0.29 |
| Ours | Ensemble | **91.5**$_{\uparrow 2.0}$ | **0.60**$_{\uparrow 0.04}$ | **92.0**$_{\uparrow 0.5}$ | **0.61**$_{\uparrow 0.04}$ | **74.5**$_{\uparrow 8.5}$ | **0.43**$_{\uparrow 0.02}$ | **86.5**$_{\uparrow 6.5}$ | **0.52**$_{\uparrow 0.04}$ | **54.5**$_{\uparrow 4.0}$ | **0.31**$_{\uparrow 0.03}$ | **52.0**$_{\uparrow 0.5}$ | **0.29**$_{\uparrow 0.00}$ |

*Table 2.* Performance comparison on representative closed-source MLLMs. Green subscripts denote gains over FOA-Attack.

| Method | Model | GPT-4o | | GPT-4.1 | | Claude-3.5 | | Claude-3.7 | | Gemini-2.0 | |
|---|---|---|---|---|---|---|---|---|---|---|---|
| | | ASR | AvgSim | ASR | AvgSim | ASR | AvgSim | ASR | AvgSim | ASR | AvgSim |
| AttackVLM | VIT-B/16 | 1.5 | 0.03 | 1.5 | 0.01 | 0.5 | 0.01 | 2.0 | 0.03 | 1.0 | 0.02 |
| | VIT-B/32 | 2.0 | 0.02 | 1.0 | 0.02 | 1.0 | 0.02 | 2.0 | 0.02 | 1.0 | 0.01 |
| | Laion | 2.5 | 0.02 | 1.5 | 0.02 | 0.5 | 0.02 | 2.0 | 0.02 | 1.5 | 0.02 |
| AdvDiffVLM | Ensemble | 4.5 | 0.04 | 5.0 | 0.04 | 2.0 | 0.02 | 3.5 | 0.03 | 3.0 | 0.03 |
| SSA-CWA | Ensemble | 2.0 | 0.03 | 2.0 | 0.02 | 2.0 | 0.02 | 2.5 | 0.03 | 3.0 | 0.03 |
| AnyAttack | Ensemble | 12.5 | 0.07 | 10.5 | 0.06 | 10.0 | 0.06 | 12.5 | 0.08 | 11.0 | 0.07 |
| M-Attack | Ensemble | 81.0 | 0.46 | 85.5 | 0.50 | 19.5 | 0.11 | 25.0 | 0.14 | 74.0 | 0.41 |
| FOA-Attack | Ensemble | 85.5 | 0.50 | 88.5 | 0.55 | 17.0 | 0.10 | 23.5 | 0.15 | 74.0 | 0.43 |
| Ours | Ensemble | **89.5**$_{\uparrow 4.0}$ | **0.53**$_{\uparrow 0.03}$ | **89.0**$_{\uparrow 0.5}$ | **0.57**$_{\uparrow 0.02}$ | **20.0**$_{\uparrow 3.0}$ | **0.12**$_{\uparrow 0.02}$ | **32.0**$_{\uparrow 8.5}$ | **0.18**$_{\uparrow 0.03}$ | **80.5**$_{\uparrow 6.5}$ | **0.47**$_{\uparrow 0.04}$ |

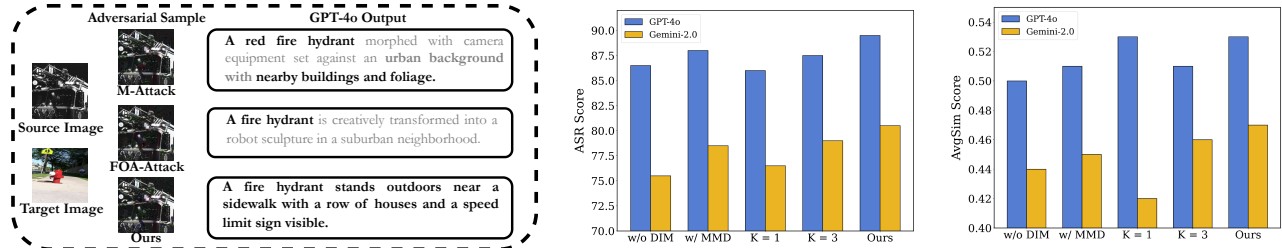

*Figure 3.* Analysis of the DART framework's effectiveness against adversarial attacks and its key components. **(Left)** Comparison of adversarial samples generated by DART and other attack methods (M-Attack, FOA-Attack) on GPT-4o. DART successfully induced the model to generate descriptions aligned with the target image's semantics, whereas other methods failed. **(Middle & Right)** Ablation studies for the DART framework are conducted. "Ours" denotes the full model with default settings ($K$=2, Energy Distance).

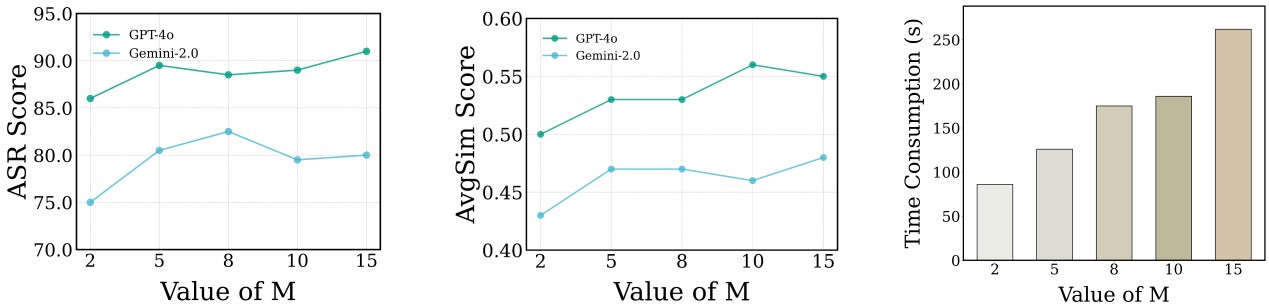

*Figure 4.* Trade-off between performance and computational cost with respect to the number of augmented samples ($M$). These results justify selecting $M$=5 for the main experiments.

DART achieved an ASR of 91.5%, surpassing FOA-Attack by 1.0%. Even on the highly robust Claude-Sonnet-4, where other methods falter, DART achieved an ASR of 14.5%, nearly 5% higher than FOA-Attack. These results show that

advanced reasoning capabilities cannot fully mitigate vulnerabilities inherited from visual encoders, whereas DART's semantic alignment strategy remains highly effective. To further assess fine-grained semantic alignment, we performed

*Table 3.* Performance comparison on closed-source MLLMs with enhanced reasoning capabilities. Green subscripts: gain over FOA.

| Method | o3 | | Claude-3.7-thinking | | Claude-Sonnet-4 | |
|---|---|---|---|---|---|---|
| | ASR | AvgSim | ASR | AvgSim | ASR | AvgSim |
| M-Attack | 87.0 | 0.50 | 23.0 | 0.14 | 12.5 | 0.08 |
| FOA-Attack | 90.5 | 0.53 | 26.5 | 0.15 | 9.5 | 0.07 |
| Ours | **91.5**$_{\uparrow 1.0}$ | **0.59**$_{\uparrow 0.06}$ | **32.0**$_{\uparrow 5.5}$ | **0.17**$_{\uparrow 0.02}$ | **14.5**$_{\uparrow 5.0}$ | **0.08**$_{\uparrow 0.01}$ |

*Table 4.* Robustness evaluation on GPT-4o target.

| Method | Original Judge | | Claude Judge | |
|---|---|---|---|---|
| | ASR | AvgSim | ASR | AvgSim |
| FOA-Attack | 85.5 | 0.50 | 79.0 | 0.51 |
| Ours | **89.5**$_{\uparrow 4.0}$ | **0.53**$_{\uparrow 0.03}$ | **84.0**$_{\uparrow 5.0}$ | **0.57**$_{\uparrow 0.06}$ |

*Table 5.* Efficiency comparison on GPT-4o. Peak VRAM is identical for $M=2$ and $M=5$, showing the extra memory comes from the GNN Router infrastructure rather than distribution estimation.

| Method | Setting | ASR (%) | Time (s) | VRAM (GB) | Speed |
|---|---|---|---|---|---|
| FOA-Attack | Std. | 85.5 | 109 | **7.8** | 1.0× |
| Ours | $M=5$ | **89.5** | 175 | 18.9 | 0.6× |
| Ours | $M=2$ | 86.0 | **86** | 18.9 | **1.3×** |

a Keyword Matching Rate (KMR) analysis. DART achieved significantly better performance than baseline models, especially in matching multiple keywords (KMR/2 and KMR/3), highlighting its strong ability to capture core semantics (see Appendix F for details).

**Case Study.** Figure 3 (Left) illustrates a challenging scenario characterized by a significant semantic gap. The baseline methods produce object hallucinations due to overfitting of the surrogate model, whereas DART generates a description that aligns closely with the target scene. This success is attributed to our dual strategy: the GNN Attack Router selects the optimal surrogate model, and distribution-aware mining enables holistic, statistically robust semantic transfer that captures both primary objects and fine-grained environmental details.

### 4.3. Ablation Study

We conducted a series of ablation studies to assess the contribution of each key component and hyperparameter in DART. First, we analyzed the sensitivity to the number of augmented samples, $M$, which is crucial for estimating the latent distribution. Figure 4 presents this analysis. The left and middle panels show that, for GPT-4o and Gemini-2.0, increasing $M$ generally improves the ASR and the AvgSim. As $M$ increases from 1 (the point-to-point baseline; see Table 6) to 5, the ASR on GPT-4o increases significantly from 83.0% to 89.5%, while AvgSim also improves from 0.51 to 0.53. The largest improvement occurs when transitioning from $M=1$ to $M=2$, where the ASR increases by

*Table 6.* Controlled ablation isolating distribution-level alignment versus pure gradient smoothing on GPT-4o. *AvgCosine* removes the intra-distribution terms from Equation 1, making it mathematically equivalent to EoT, while sharing all other components with DART.

| Variant | Loss Function ($M$) | ASR (%) | AvgSim |
|---|---|---|---|
| Point-wise (no aug.) | Cosine ($M=1$) | 83.0 | 0.51 |
| EoT-equivalent | AvgCosine ($M=5$) | 82.5 | 0.45 |
| DART (Ours) | Energy Distance ($M=5$) | **89.5** | **0.53** |

3.0%. This phenomenon occurs because, when $M=1$, the Energy Distance degenerates into a point-to-point matching scheme, causing the intra-distribution terms $\mathbb{E}[d_c(\tilde{X}, \tilde{X}')]$ and $\mathbb{E}[d_c(\tilde{Y}, \tilde{Y}')]$ to vanish. Beyond $M=5$, the performance gain gradually saturates, and the ASR on GPT-4o shows only marginal improvement as $M$ further increases. Meanwhile, the figure on the right shows that the computation time per attack instance increases substantially with larger values of $M$. To balance attack effectiveness and computational efficiency, we selected $M=5$ for all experiments because it achieves near-optimal performance while maintaining moderate computational overhead.

**Distribution-level alignment vs. gradient smoothing.** To verify that the performance gain of DART arises from distribution-level alignment rather than merely from gradient smoothing induced by data augmentation, we construct an EoT-equivalent baseline, denoted as *AvgCosine*. Specifically, we remove the intra-distribution terms $\mathbb{E}[d_c(\tilde{X}, \tilde{X}')]$ and $\mathbb{E}[d_c(\tilde{Y}, \tilde{Y}')]$ from Equation 1 while keeping all other components, including five augmented samples, the GNN Router, and dynamic weighting, unchanged. As shown in Table 6, the EoT-equivalent baseline achieves an ASR of 82.5%, which does not improve upon the point-wise baseline with $M = 1$ (83.0%). This result indicates that simple gradient smoothing provides little benefit. In contrast, the full Energy Distance objective achieves an ASR of 89.5%, representing improvements of +7.0% over the EoT baseline and +6.5% over the point-wise baseline. These results confirm that the performance gain originates from the distribution-aware semantic alignment introduced by the intra-distribution calibration term, rather than from gradient smoothing alone.

Beyond increasing the sample size, we examined the effect of the weighting factor $\eta_k$, which balances global and local feature alignment. Analysis presented in Appendix E confirms that the heterogeneous weighting scheme ($\eta=[1.8, 2.1, 4.8]$) significantly outperforms the uniform scheme. These results support our hypothesis that surrogate models with varying architectures benefit from customized feature balancing.

We also evaluated other critical components of the framework, with results shown in Figure 3. Notably, the ablation study results shown in the middle and right panels of Figure

*Table 7.* Robustness of DART against common input preprocessing defenses on GPT-4o. Adversarial samples are perturbed by each defense before being fed to the target MLLM.

| Defense | FOA-Attack | Ours | $\Delta$ |
|---|---|---|---|
| No defense | 85.5 | **89.5** | +4.0 |
| JPEG ($Q$=75) | 80.0 | **87.5** | +7.5 |
| Gaussian Blur ($r$=1.0) | 53.0 | **65.0** | **+12.0** |
| Bit-depth (4-bit) | 82.5 | **84.0** | +1.5 |

3 provide several key insights. The DART variant without DIM reduces to simple point-wise feature matching, showing the most substantial performance drop. This finding supports our core hypothesis that aligning latent distributions is essential for generalization, since point-wise alignment tends to overfit the specific feature space of the surrogate model. Furthermore, we observe that Energy Distance consistently outperforms MMD, likely because MMD is highly sensitive to kernel function selection, whereas Energy Distance is non-parametric and more robust. Finally, the GNN Attack Router achieves the optimal ensemble scale at $K$=2. When $K$=1, the gradient diversity is insufficient, whereas $K$=3 results in a slight performance decline, possibly because conflicting gradients from less suitable surrogate models complicate the optimization process. Furthermore, compared with random or static integration methods, the GNN Attack Router introduces negligible inference latency while effectively capturing complex, task-dependent transfer patterns that simple heuristic methods fail to identify.

### 4.4. Robustness and Efficiency Analysis

**Robustness of Evaluation.** To avoid potential bias from using the same model family for both image description and evaluation, we performed rigorous cross-judge validation. We used Claude-Sonnet-4.5 as an independent judge. As shown in Table 4, although the absolute ASR scores varied because of different judge sensitivities, DART consistently outperformed the strongest baseline, FOA-Attack, across all evaluation protocols. Notably, under Claude-Sonnet-4.5 evaluation, the performance gap widened to 5.0%, confirming that our improvements arise from robust semantic alignment rather than judge-specific factors.

**Robustness against Input Preprocessing Defenses.** In real-world systems, lightweight input-side defenses are commonly employed to mitigate adversarial attacks. To evaluate the performance of DART under these conditions, we applied three common preprocessing defenses to the adversarial samples before feeding them into GPT-4o: JPEG compression ($Q = 75$), Gaussian blurring ($r = 1.0$), and 4-bit depth reduction. As shown in Table 7, DART performs better compared with FOA-Attack across all defense settings, and its advantage further *widens* under stronger defenses, particularly under Gaussian blurring, where the

performance gap increases from $+4.0\%$ without defense to $+12.0\%$. These results suggest that distribution-aligned perturbations capture more robust semantic features and are therefore more resilient to input transformations, whereas per-pixel perturbations are more vulnerable to such distortions.

**Computational Efficiency.** We further analyze the trade-off between attack performance and computational cost. While our default setting ($M = 5$) prioritizes maximum transferability, DART provides a flexible and efficient configuration. As shown in Table 5, even with reduced data augmentation ($M = 2$), DART remains 1.3 times faster than FOA-Attack (86 s vs. 109 s per image) while still achieving a higher ASR (86.0% vs. 85.5%). Regarding memory, the peak VRAM is identical for $M=2$ and $M=5$ (both 18.9 GB), confirming that the increase over FOA-Attack (7.8 GB) is a fixed overhead from the GNN router infrastructure rather than from distribution estimation. Together, these results indicate that the performance gain arises from optimized Energy-Distance objectives and adaptive routing, rather than from simply increasing the computational budget.

**Cross-dataset Generalization Capability.** To evaluate whether our method learns dataset-independent semantic features, we evaluated DART on the Flickr30k dataset (transferring from NIPS to Flickr30k). As reported in Appendix J, DART consistently outperforms competing methods, achieving attack success rates of 85.0% on GPT-4o (vs. 83.5% for FOA-Attack) and 77.5% on Gemini-2.0 (vs. 70.5% for FOA-Attack). These experimental results strongly indicate that the learned distribution alignment successfully captures intrinsic semantic properties that are highly robust to challenging domain shifts, rather than merely overfitting to the specific source dataset.

## 5. Conclusion

This paper introduces DART, a novel framework that statistically reconstructs adversarial attacks on closed-source MLLMs. DART addresses the limitations of traditional point-by-point feature matching through a dual-pronged strategy: 1) Distribution-aware Intrinsic Mining, which leverages Energy Distance to capture semantic manifolds in images and improve generalization; and 2) GNN Attack Router, which replaces heuristic ensembles with data-driven methods for selecting specialized surrogate models. Extensive experiments show that DART consistently outperforms state-of-the-art methods across a variety of MLLMs. This work establishes a more robust baseline for evaluating multi-modal system security and highlights the critical importance of distribution alignment. Our findings provide a general perspective on improving robustness and enhancing transferability across a wider range of cross-modal learning tasks.

## Impact Statement

This paper proposes an attack-oriented approach to improving the security of MLLMs. By revealing vulnerabilities in state-of-the-art MLLMs using DART, we aim to facilitate the development of more robust defense mechanisms for critical applications, such as autonomous driving and content moderation. To mitigate ethical risks, we rely on standard public datasets and restrict attack targets to benign semantic contexts, thereby avoiding hate speech and discriminatory content. While acknowledging the dual-use risks of adversarial attacks, we argue that disclosing these vulnerabilities is essential for building trustworthy and resilient AI systems.

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

# A. Analysis of Distribution Alignment

This section provides an explanation for why the proposed Distribution-aware Intrinsic Mining method offers better generalization than traditional point-by-point feature matching approaches, drawing on the statistical properties of Energy Distance.

## A.1. Problem Description and Limitations of Point-wise Matching

Let $x$ denote the input image, and let $G_\theta(\cdot)$ represent the surrogate model's global feature extractor (as defined in Section 3.2). Previous methods (e.g., M-Attack and FOA-Attack) perform point-wise alignment and treat the semantic target as a deterministic vector, $z_{tar} = G_\theta(x_{tar})$. The optimization objective is as follows:

$$\min_\delta \|G_\theta(x_{nat} + \delta) - G_\theta(x_{tar})\|^2. \tag{3}$$

However, $G_\theta(x_{tar})$ provides only one realization of the target semantics. In high-dimensional latent spaces, this isolated point is easily perturbed by high-frequency artifacts or model-specific biases, which can cause overfitting to particular curvatures of the empirical risk minimization surrogate.

## A.2. Energy Distance as a Strictly Proper Scoring Rule

DART treats the target embedding as a random variable $Z_{tar} \sim \mathcal{D}_G(x_{tar})$ and approximates it empirically using $M$ augmented views. We adopt the Energy Distance ($D_E$), a statistically robust metric based on the distance between distribution characteristic functions. Crucially, the Energy Distance is a strictly proper scoring rule (Gneiting & Raftery, 2007; Rizzo & Székely, 2016), with the asymptotic identifiability property:

$$D_E^2(P, Q) = 0 \iff P = Q, \tag{4}$$

where $P$ and $Q$ denote the distributions of adversarial features and target global features, respectively. In practice, we approximate the target distribution using a finite set of $M$ augmented samples in a high-dimensional latent space, thereby enabling distribution-aware semantic alignment under finite-sample conditions rather than exact distribution matching. Even under this approximation, the strict properness of this scoring rule ensures that the optimization process guides perturbations toward the true semantic manifold rather than toward isolated surrogate points.

## A.3. Gradient Consistency and Variance Reduction

In our global alignment, we employ a point-to-distribution strategy in which the adversarial image feature $g_{adv} = G_\theta(x_{adv})$ is modeled as a deterministic point (Dirac measure), whereas the target $Z_{tar}$ is assumed to follow a distribution. In this setting, the Energy Distance objective function simplifies to:

$$\mathcal{L}_{DART} \propto \mathbb{E}_{z \sim \mathcal{D}_G(x_{tar})}[\|G_\theta(x_{adv}) - z\|]. \tag{5}$$

Using a batch of $M$ augmented target samples, the gradient estimate becomes:

$$\nabla_\delta \mathcal{L}_{DART} \approx \nabla_\delta \left( \frac{1}{M} \sum_{i=1}^{M} \|G_\theta(x_{adv}) - z_i\| \right). \tag{6}$$

Compared with point-wise matching, this formulation acts as a variance-reduction technique for gradient estimators. By averaging out the high-frequency noise introduced by individual augmented views, the optimized trajectory becomes smoother, which helps prevent adversarial perturbations from being trapped in sharp, model-specific local minima. This theoretical insight directly accounts for the observed improvement in transferability across diverse black-box architectures.

## A.4. Role of Intra-Distribution Terms: Calibration without Divergence

We analyze the intra-distribution terms $\mathbb{E}[d_c(\tilde{X}, \tilde{X}')]$ and $\mathbb{E}[d_c(\tilde{Y}, \tilde{Y}')]$ in Equation 1 to demonstrate that they neither introduce semantic divergence in the global branches nor cause feature collapse in the local branches.

**Global branch ($p \rightarrow d$): the intra-term vanishes.** On the adversarial side of global alignment, the feature $g_{adv} = G_\theta(x_{adv})$ is represented as a single deterministic vector. If this vector is regarded as the Dirac measure $\delta_{g_{adv}}$, then the two samples $\tilde{X}$

and $\tilde{X}'$ are almost surely identical, yielding $\mathbb{E}[d_c(\tilde{X}, \tilde{X}')] \equiv 0$. The target-side intra-distribution term, $\mathbb{E}[d_c(\tilde{Y}, \tilde{Y}')]$, is a constant determined by the augmented target views and is independent of $\delta$. Consequently, the gradient with respect to $\delta$ is contributed solely by the cross-distribution term $\mathbb{E}[d_c(\tilde{X}, \tilde{Y})]$, while no distribution-internal maximization exists in the global branch.

**Local branch ($d \rightarrow d$): the intra-term acts as calibration.** For local features, both the adversarial and target domains are modeled as empirical distributions over the $K$-means cluster centers. In this setting, the intra-distribution terms influence the loss surface by maintaining the strictly proper scoring property. Without these terms, a degenerate solution in which all adversarial cluster centers collapse to a single point could still trivially minimize the cross-distribution terms. Therefore, the intra-distribution terms serve as a calibration mechanism that penalizes such collapses and encourages the adversarial cluster centers to maintain a dispersion comparable to that of the target centers.

**Empirical verification.** We tracked three quantities over 300 optimization steps: the global-branch cross-distribution metric $E_{\text{cross}}(G)$, the local-branch adversarial-side intra-distribution metric $E_{\text{intra}}(L_{adv}) = \mathbb{E}[d_c(\tilde{X}, \tilde{X}')]$, and the local-branch target-side intra-distribution metric $E_{\text{intra}}(L_{tar}) = \mathbb{E}[d_c(\tilde{Y}, \tilde{Y}')]$. As shown in Table 8, $E_{\text{intra}}(L_{adv})$ remains stable at approximately $0.577$ throughout the optimization process, closely matching the target's intrinsic variance $E_{\text{intra}}(L_{tar}) \approx 0.59$, which empirically confirms that the adversarial-side intra-distribution neither diverges nor collapses. Meanwhile, $E_{\text{cross}}(G)$ decreases steadily ($0.671 \rightarrow 0.233$) and the total loss drops from $5.87$ to $1.74$, indicating that the global cross-distance is being effectively minimized while the intra-distribution stays calibrated. The intra-distribution term therefore behaves as a calibration mechanism rather than as a source of divergence.

*Table 8.* Tracking the global cross-distribution and local intra-distribution Energy-Distance terms during adversarial optimization. $E_{\text{intra}}(L_{adv})$ remains stable around the target's intrinsic variance $E_{\text{intra}}(L_{tar}) \approx 0.59$, while $E_{\text{cross}}(G)$ steadily decreases.

| Step | $E_{\text{cross}}(G)$ | $E_{\text{intra}}(L_{adv})$ | $E_{\text{intra}}(L_{tar})$ | Total Loss |
|---|---|---|---|---|
| 0 | 0.671 | 0.575 | 0.576 | 5.87 |
| 100 | 0.515 | 0.577 | 0.594 | 3.81 |
| 300 | 0.233 | 0.578 | 0.594 | 1.74 |

# B. Offline Training of the GNN Attack Router

## B.1. Network Architecture Implementation

To implement the GNN Attack Router, we adopt a heterogeneous GraphSAGE architecture built on the PyTorch Geometric library.

**Encoder.** The encoder comprises two layers of heterogeneous graph convolutions (HeteroConv). Each layer uses a SAGEConv operator with mean pooling to aggregate information from neighboring nodes. The hidden feature dimension is set to 128.

**Predictor.** The edge predictor is a two-layer MLP (Linear $\rightarrow$ ReLU $\rightarrow$ Linear) that concatenates the updated features of task and model nodes to predict scalar transfer scores.

**Input Projection.** A linear projection layer maps the high-dimensional task features (ViT embeddings) and model features (one-hot encoded) onto a shared hidden space ($d = 128$) before they are fed into the GNN encoder.

## B.2. Offline GNN Attack Router Training

This appendix describes the offline training process for the GNN Attack Router, as outlined in Algorithm 1. The process begins by constructing a heterogeneous graph from historical attack logs and pre-extracted image features. The graph consists of "attack_task" nodes and "base_model" nodes. Edges between nodes represent attack evaluations and are labeled with "transfer_score".

The GNN is trained with link prediction to estimate "transfer_score", enabling it to identify effective attack routes. We utilize the Adam optimizer with a learning rate of $1 \times 10^{-3}$ and an MSE loss, and prevent overfitting via early stopping. The best-performing model is preserved for the online routing phase.

---

**Algorithm 1** Offline GNN Attack Router Training

---

**Require:** Attack logs $L$, Source features $\{F_s\}$, Target features $\{F_t\}$
**Ensure:** Trained GNN Router model parameters $\theta^*$

 1: **function** TRAIN_ATTACK_ROUTER($L, \{F_s\}, \{F_t\}$)
 2:     Initialize a heterogeneous graph $G$
 3:     Construct node features $G.\mathcal{V}_{task}.X$ and $G.\mathcal{V}_{model}.X$ from $\{F_s\}, \{F_t\}$ and logs $L$.
 4:     Construct edges $E_{index}$ and labels $Y$ from logs $L$.
 5:     Initialize GNN model $\mathcal{M}(\theta)$, optimizer $\mathcal{O}$, and loss $\mathcal{L} \leftarrow$ MSE Loss.
 6:     Split edges and labels into training $(E_{train}, Y_{train})$ and validation $(E_{val}, Y_{val})$ sets.
 7:     $Y'_{train} \leftarrow Y_{train}$
 8:     $Y'_{val} \leftarrow Y_{val}$
 9:     $loss_{best\_val} \leftarrow \infty, \theta^* \leftarrow \theta$
10:     **for** epoch = $1 \rightarrow E_{max}$ **do**
11:         $\mathcal{M}$.train()
12:         $\hat{Y}_{train} \leftarrow \mathcal{M}(G.X, E_{train})$ {Forward pass on training subgraph}
13:         $loss_{train} \leftarrow \mathcal{L}(\hat{Y}_{train}, Y'_{train})$
14:         Update $\theta$ using backpropagation on $loss_{train}$.
15:         **if** epoch % 10 == 0 **then**
16:             $\mathcal{M}$.eval()
17:             $\hat{Y}_{val} \leftarrow \mathcal{M}(G.X, E_{val})$
18:             $loss_{val} \leftarrow \mathcal{L}(\hat{Y}_{val}, Y'_{val})$
19:             **if** $loss_{val} < loss_{best\_val}$ **then**
20:                 $loss_{best\_val} \leftarrow loss_{val}, \theta^* \leftarrow \theta$ {Save best model}
21:             **end if**
22:         **end if**
23:         Check for early stopping condition.
24:     **end for**
25:     **return** $\theta^*$
26: **end function**

---

## C. Online Adversarial Attack Generation

This appendix provides pseudocode (Algorithm 2) for the online attack generation phase and details how to craft adversarial examples for a new task using the pre-trained GNN Attack Router and Distribution-aware Intrinsic Mining.

The process starts from a new attack task defined by a source image $x_{nat}$ and a target image $x_{tar}$. First, the system queries the pre-trained GNN Attack Router (see Algorithm 1) to predict the transferability of all available surrogate models for the task. It then selects the Top-$K$ models to form a task-adaptive ensemble. The algorithm then enters an iterative optimization loop. Inside the loop, it estimates the intrinsic latent distribution of the target image by applying several data augmentations. The optimizer minimizes the Energy Distance between the current adversarial sample and the target's latent distribution, evaluated over the selected ensemble. Perturbations are updated by gradient-based methods and projected onto the $\ell_p$-norm ball to ensure imperceptibility. This process repeats for a fixed number of steps to produce the final adversarial sample.

## D. Additional Implementation Details

Following prior work (Li et al., 2025; Jia et al., 2025), we use three CLIP architectures as surrogates: ViT-B/16, ViT-B/32, and ViT-g-14-Laion-2B-s12B-b42K (Ilharco et al., 2021). The perturbation budget is $\epsilon = 16/255$, step size $\alpha = 1/255$, and $\tau = 300$ iterations. The GNN Attack Router uses task node features derived from ViT-g-14-Laion-2B-s12B-b42K embeddings. All experiments were conducted on a single NVIDIA A800 GPU with PyTorch. Attack Success Rate (ASR) is the proportion of test samples whose semantic similarity score exceeds 0.3, following Li et al. (2025). Average Similarity Score (AvgSim) is the mean semantic similarity between generated and target descriptions. Keyword Matching Rate KMR/1, KMR/2, and KMR/3 denote the proportions of descriptions that contain one, two, and three manually assigned target keywords, respectively. For the KMR analysis, we used the 100-pair dataset configuration proposed by Li et al. (2025). To avoid data leakage, we excluded datasets overlapping with the main training set and retrained the GNN Attack Router on the remaining 700-pair dataset.

## E. Hyperparameter Analysis of $\eta_k$

In Section 3.2, the hyperparameter $\eta_k$ acts as a key weighting factor that balances the contributions of global features ($\mathcal{L}_{global}$) and local features ($\mathcal{L}_{local}$) for each surrogate model $k$. The expressive power of global and local features varies across model architectures. Therefore, identifying an appropriate balance is crucial for effective feature alignment. We evaluated the sensitivity of our framework under different settings of $\eta_k$.

As shown in Table 9, we empirically selected heterogeneous weights for ViT-B/16, ViT-B/32, and Laion models ($\eta$ = [1.8, 2.1, 4.8], respectively) and compared them with several uniform settings where all models share identical weights ([1,1,1], [3,3,3], [5,5,5]). The results demonstrate that uniform weights consistently yield suboptimal performance. Our proposed non-uniform weighting scheme achieves the highest ASR and AvgSim on both the GPT-4o and Gemini-2.0 models. These results confirm our hypothesis that assigning distinct weights based on each surrogate model's characteristics is a more effective strategy, and they also validate the rationale behind the $\eta_k$ values chosen in our main experiments.

*Table 9.* The effect of the weighting factor $\eta_k$ on attack performance.

| Value of $\eta_k$ | GPT-4o | | Gemini-2.0 | |
|---|---|---|---|---|
| | ASR | AvgSim | ASR | AvgSim |
| $[1, 1, 1]$ | 86.5 | 0.52 | 73.5 | 0.44 |
| $[3, 3, 3]$ | 87.5 | 0.52 | 80.0 | 0.44 |
| $[5, 5, 5]$ | 86.0 | 0.50 | 75.5 | 0.43 |
| $[1.8, 2.1, 4.8]$ | **89.5** | **0.53** | **80.5** | **0.47** |

## F. Detailed KMR Analysis

To assess semantic alignment at a finer level, we used the KMR metric. Table 10 shows that DART produces adversarial descriptions that are more semantically aligned with the target. On GPT-4o, DART achieved KMR/1, KMR/2, and KMR/3

---

**Algorithm 2** Online DART Attack Generation

---

**Require:** Source image $x_{nat}$, Target image $x_{tar}$
 1: Pre-trained GNN Attack Router $\mathcal{M}_{\theta^*}$
 2: Full set of surrogate models $\mathcal{F} = \{f_1, f_2, ..., f_N\}$
 3: Feature extractor for GNN input $f_{feat}$
 4: Attack parameters: perturbation budget $\epsilon$, step size $\alpha$, number of steps $T_{steps}$
 5: Ensemble size $K$, Number of augmentations $M$
**Ensure:** Adversarial image $x_{adv}$

 6: **procedure** DART_ATTACK($x_{nat}, x_{tar}, \mathcal{M}_{\theta^*}, \mathcal{F}, f_{feat}, \epsilon, \alpha, T_{steps}, K, M$)
 7:     {**Phase 1: Adaptive Surrogate Model Selection**}
 8:     $feat_{nat} \leftarrow f_{feat}(x_{nat})$
 9:     $feat_{tar} \leftarrow f_{feat}(x_{tar})$
10:     $feat_{task} \leftarrow$ CONCAT($feat_{nat}, feat_{tar}$) {Construct task node feature}
11:     $Scores \leftarrow \mathcal{M}_{\theta^*}(feat_{task}, \mathcal{F})$ {Predict transfer scores for all models}
12:     $\mathcal{F}^* \leftarrow$ Top-K($Scores, K$) {Select the best $K$ surrogate models}

13:     {**Phase 2: Iterative Adversarial Optimization**}
14:     $\delta \leftarrow \mathbf{0}$ {Initialize perturbation}
15:     $x_{adv} \leftarrow x_{nat} + \delta$
16:
17:     {Estimate target's intrinsic distributions}
18:     $\{x'_{tar,i}\}_{i=1}^M \leftarrow \{\text{Augment}_i(x_{tar})\}_{i=1}^M$ {Apply $M$ random augmentations}
19:     For each model $f_k \in \mathcal{F}^*$:
20:         $\mathcal{D}_G^{(k)}(x_{tar}) \leftarrow \{G_{\theta_k}(x'_{tar,i})\}_{i=1}^M$ {Compute global feature distribution}
21:         $\mathcal{D}_L^{(k)}(x_{tar}) \leftarrow \{\mathcal{C}(L_{\theta_k}(x'_{tar,i}))\}_{i=1}^M$ {Compute local feature distribution}
22:
23:     **for** $t = 1 \rightarrow T_{steps}$ **do**
24:         $\mathcal{L}_{total} \leftarrow 0$
25:         For each model $f_k \in \mathcal{F}^*$:
26:             $g_{adv}^{(k)} \leftarrow G_{\theta_k}(x_{adv})$ {Extract global feature of adv sample}
27:             $C_{adv}^{(k)} \leftarrow \mathcal{C}(L_{\theta_k}(x_{adv}))$ {Extract local feature clusters of adv sample}
28:             $\mathcal{L}_{global}^{(k)} \leftarrow E_{p \rightarrow d}(g_{adv}^{(k)}, \mathcal{D}_G^{(k)}(x_{tar}))$ {Point-to-dist Energy Distance}
29:             $\mathcal{L}_{local}^{(k)} \leftarrow \frac{1}{M} \sum_{i=1}^M E_{d \rightarrow d}(C_{adv}^{(k)}, \mathcal{D}_L^{(k)}(x_{tar})_i)$ {Avg dist-to-dist Energy Distance}
30:             $\mathcal{L}_{align}^{(k)} \leftarrow \mathcal{L}_{global}^{(k)} + \eta_k \mathcal{L}_{local}^{(k)}$
31:         end for
32:
33:         {Calculate dynamically weighted total loss}
34:         $\{w_k\}_{k=1}^K \leftarrow$ DynamicWeights($\{\mathcal{L}_{align}^{(k)}\}_{k=1}^K$)
35:         $\mathcal{L}_{total} \leftarrow \sum_{k=1}^K w_k \cdot \mathcal{L}_{align}^{(k)}$
36:
37:         $g \leftarrow \nabla_\delta \mathcal{L}_{total}$ {Compute gradient w.r.t. perturbation}
38:         $\delta \leftarrow \delta - \alpha \cdot \text{sign}(g)$ {Update perturbation}
39:         $\delta \leftarrow \text{clip}(\delta, -\epsilon, \epsilon)$ {Project perturbation into $\ell_\infty$-ball}
40:         $x_{adv} \leftarrow \text{clip}(x_{nat} + \delta, 0, 1)$ {Ensure valid image range}
41:     **end for**
42:
43:     **return** $x_{adv}$
44: **end procedure**

---

scores of 0.90, 0.71, and 0.25, respectively, consistently surpassing all baseline methods. The substantial improvements in KMR/2 and KMR/3—which involve matching multiple keywords—clearly indicate that DART's distribution alignment captures the core semantics of target images more effectively than point-by-point feature matching. A similar trend appears on Gemini-2.0, where DART increases KMR/3 to 0.20—almost twice that of the second-best method.

*Table 10.* KMR comparison with representative state-of-the-art models. Green subscripts denote gains over FOA-Attack.

| Method | Model | GPT-4o | | | Gemini-2.0 | | |
|---|---|---|---|---|---|---|---|
| | | KMR/1 | KMR/2 | KMR/3 | KMR/1 | KMR/2 | KMR/3 |
| AttackVLM | VIT-B/16 | 0.09 | 0.04 | 0.00 | 0.07 | 0.02 | 0.00 |
| | VIT-B/32 | 0.08 | 0.02 | 0.00 | 0.06 | 0.02 | 0.00 |
| | Laion | 0.07 | 0.04 | 0.00 | 0.07 | 0.02 | 0.00 |
| AdvDiffVLM | Ensemble | 0.02 | 0.00 | 0.00 | 0.01 | 0.00 | 0.00 |
| SSA-CWA | Ensemble | 0.11 | 0.06 | 0.00 | 0.05 | 0.02 | 0.00 |
| AnyAttack | Ensemble | 0.44 | 0.20 | 0.04 | 0.46 | 0.21 | 0.05 |
| M-Attack | Ensemble | 0.82 | 0.54 | 0.13 | 0.75 | 0.53 | 0.11 |
| FOA-Attack | Ensemble | 0.88 | 0.64 | 0.25 | 0.77 | 0.56 | 0.12 |
| Ours | Ensemble | $\mathbf{0.90}_{\uparrow 0.02}$ | $\mathbf{0.71}_{\uparrow 0.07}$ | $\mathbf{0.25}_{\uparrow 0.00}$ | $\mathbf{0.83}_{\uparrow 0.06}$ | $\mathbf{0.60}_{\uparrow 0.04}$ | $\mathbf{0.20}_{\uparrow 0.08}$ |

# G. Evaluation Prompts

For both ASR/AvgSim and KMR evaluation, we adopt the same prompts as Li et al. (2025). We refer readers to Li et al. (2025) for the detailed prompt content.

# H. More Examples of Attack Effect Visualizations

To offer a clearer and more comprehensive demonstration of the performance and robustness of the proposed DART framework against MLLMs, this appendix presents five additional qualitative visualization examples. These examples complement Figure 1 in the main text and further demonstrate, from multiple perspectives, that DART-generated adversarial samples can effectively deceive a range of advanced MLLMs. This deception leads MLLMs to produce descriptions aligned with the semantic content of the target image, rather than the actual content of the adversarial sample. We deliberately selected examples across diverse themes, levels of complexity, and types of challenges to comprehensively evaluate the DART framework's generalization ability and attack precision.

As shown in Figure 5, this case highlights DART's strong performance in processing image pairs that contain two visually unrelated concepts. The adversarial sample is a photograph of a sailboat, whereas the target semantic comes from a top-down image featuring stationery, backpacks, and electronic devices. Despite large differences in subject matter, color, composition, and underlying concepts, all tested models ignored the prominent sailboat and instead produced descriptions closely aligned with the target content, such as "tools" and "office supplies." This demonstrates that DART's distribution alignment strategy effectively bridges wide semantic gaps, enabling robust semantic transfer.

Figure 6 illustrates DART's ability to adjust the visual focus of models in comparable scenes. The adversarial sample shows a tram on an urban street, whereas the target sample presents a close-up of a fruit stall. Both fall under the category of urban landscapes, but their focal subjects differ completely. Experimental results show that DART effectively guided all models to disregard the large tram occupying a central position in the scene. Instead, the models carefully described the target image with terms such as "fruit market," "apples and oranges," and "price tags." This demonstrates that DART not only achieves broad semantic transfer but also provides precise, fine-grained control over model attention in complex scenes.

Figure 7 illustrates DART's ability to precisely alter key scene attributes in images, such as weather conditions and activity types. The adversarial sample shows people cycling on a park path under overcast skies, but without snow. The target sample depicts a cross-country skiing competition in snowy winter conditions. All model-generated descriptions explicitly referenced target elements such as "cross-country skiing," embedding the core attributes of "snow" and "skiing"—features entirely absent from the adversarial sample. This shows that DART effectively manipulates a model's perception of key scene attributes, rather than simply substituting the subject.

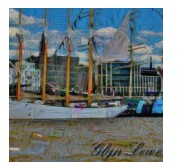

**Adversarial Sample**

**GPT-4o**    Describe this image.
A variety of household items and tools are neatly organized on a textured blue blanket.

**Claude-3.7**   Describe this image.
Colorful pencils arranged on a map with a plastic bag and notebook visible above.

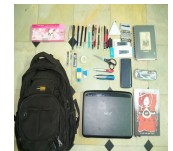

**Target Sample**

**o3**    Describe this image.
An assortment of stationery and small gadgets is neatly arranged on a patterned blanket backdrop.

**Gemini-2.0**   Describe this image.
The image shows an oddly arranged collection of office supplies and personal items against a fabric backdrop.

*Figure 5.* Transfer attacks across large semantic gaps.

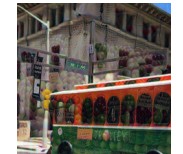

**Adversarial Sample**

**GPT-4o**    Describe this image.
A colorful fruit market stall displays an assortment of apples and prices on chalkboard signs.

**Claude-3.7**   Describe this image.
A colorful market stall displaying various products, likely food items, with signage and price tags visible.

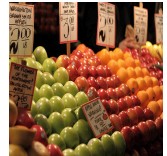

**Target Sample**

**o3**    Describe this image.
Colorful fruit stand with stacked apples and oranges brightens a busy city street market.

**Gemini-2.0**   Describe this image.
A fruit and vegetable stand is selling a variety of produce in an outdoor market.

*Figure 6.* Focus manipulation attacks in similar scenarios.

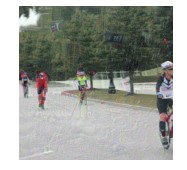

**Adversarial Sample**

**GPT-4o**    Describe this image.
Four athletes compete in a snowy cross-country skiing race amidst a wintry forest backdrop.

**Claude-3.7**   Describe this image.
Skiers in colorful jackets training or racing on a cross-country ski course through a wooded area.

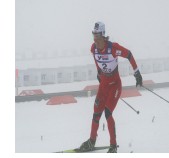

**Target Sample**

**o3**    Describe this image.
Several athletes compete in a snowy cross-country skiing race, pushing forward with poles amid falling snow.

**Gemini-2.0**   Describe this image.
Competitors are participating in a winter biathlon event, skiing and preparing to shoot at the range.

*Figure 7.* Tampering attacks on critical scene attributes.

The challenge shown in Figure 8 arises from the model's need to focus on describing a more complex human-object interaction. Results indicate that all models successfully focused their descriptions on the "teddy bear" and the person holding it. This demonstrates that DART attacks are highly effective, capable of suppressing prominent visual features in adversarial samples and reconstructing a more detailed scene narrative according to the target distribution.

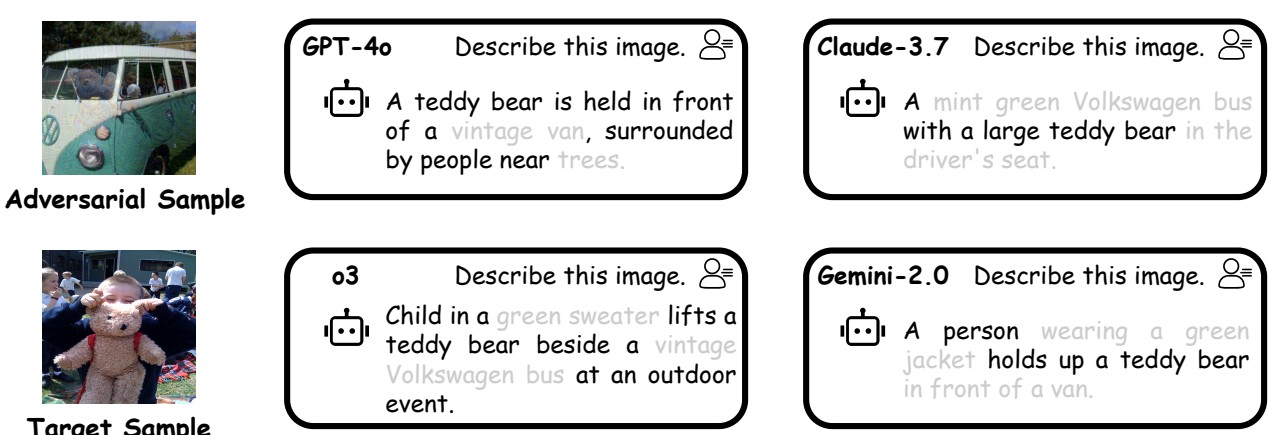

*Figure 8.* Subject replacement attack.

Figure 9 illustrates DART's robustness against adversarial attacks under visual distortion. The adversarial sample displayed an inverted traffic light, whereas the target sample showed a normal traffic light along a coastal highway. The models initially classified the image as inverted, but their detailed description later fully incorporated the target sample's scene information, including "distant hills and greenery" and a "scenic coastal view." This indicates that DART's learned latent distribution has strong semantic coherence. Even when the input image's geometric structure is distorted, the model is compelled to correct and refine its scene understanding using the target's semantic information, highlighting the robust resilience of the DART framework.

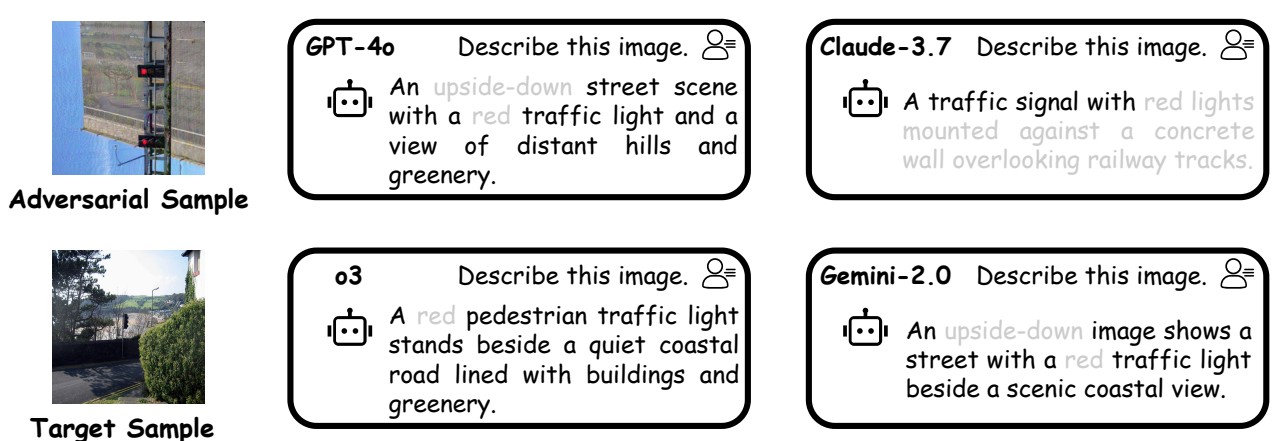

*Figure 9.* Effective attacks under visual distortions.

# I. Limitations and Future Work

Although the DART framework has achieved state-of-the-art results on multiple benchmarks, it still has certain limitations. This appendix discusses these limitations and outlines potential future directions.

## I.1. Adaptability and Scalability

The current framework encounters two major challenges: adaptability and scalability (Zhao et al., 2023). The first challenge is its reliance on static historical data, as the GNN Attack Router's high performance depends on offline pre-training with large-scale historical attack logs. This reliance restricts the framework's adaptability to new tasks, creating an adaptability bottleneck in the context of rapid model iteration. The second challenge lies in the computational overhead of distributional alignment, as the effectiveness of Distribution-aware Intrinsic Mining depends heavily on the number $M$ of data augmentation samples used for distribution estimation. Experiments show that although larger $M$ values improve attack success rates, they also increase computational costs linearly. This trade-off creates scalability challenges in scenarios demanding rapid attack generation or large-scale deployment.

To improve the framework's adaptability and scalability, we plan to explore lightweight and online adaptation mechanisms in future work. For example, transfer-learning prediction methods based on meta-learning or zero-shot learning could allow the GNN Attack Router to quickly adapt to new models while reducing dependence on historical data. At the same time, exploring more efficient distribution estimation techniques—such as using parameterized models instead of empirical sampling—could substantially reduce computational overhead while maintaining performance, thereby improving the framework's overall scalability.

## I.2. Application Scope and Method Design

The current design of DART requires further improvement in both application scope and methodological completeness. First, the digital threat model remains limited. Similar to most existing studies, our work primarily evaluates attack performance under $\ell_p$-norm constraints in the digital domain. The effectiveness of this framework against more challenging physical-world attacks remains unverified. Complex physical-world transformations may distort the estimated potential distributions, thereby reducing the direct applicability of current methods. Second, the framework still contains residual heuristic components. Although DART aims to replace manual rules with data-driven approaches, its design still incorporates heuristic choices, including the type of data augmentation transformation used for distribution estimation and the weighting factor $\eta_k$ that balances global and local feature losses. Such fixed hyperparameters may not be optimal across different tasks and models.

In the future, we will expand the application scope of DART and enhance its automated design. A key direction is to extend the framework to the physical world by incorporating physical transformations into the distribution estimation process, which will produce adversarial samples more resilient to real-world conditions. Methodologically, we will investigate learnable automation modules, such as using reinforcement learning to discover optimal data augmentation strategies, to reduce dependence on heuristic manual design. This will further improve the framework's versatility and efficiency.

## I.3. Reliance on CLIP-based Surrogate Encoders

A further limitation lies in the homogeneity of our surrogate model pool. To ensure a strictly fair comparison with prior work (Li et al., 2025; Jia et al., 2025), our experiments rely exclusively on three CLIP-based encoders (ViT-B/16, ViT-B/32, and ViT-g-14-Laion-2B-s12B-b42K). While these models differ in scale and pre-training data, they share a common contrastive image–text training paradigm and therefore expose adversarial perturbations to fundamentally similar feature processing pipelines. Whether DART's distribution-aware alignment continues to hold for surrogate encoders trained under fundamentally different paradigms—such as SigLIP, DINOv2, or diffusion-based vision encoders—remains an open question and constitutes an important stress test of the framework's true cross-encoder generality. We view extending the surrogate pool to such architecturally diverse encoders as a critical next step, which we plan to systematically investigate in future work.

## I.4. Benchmark Scope and Evaluation Bias

Although we validated our method on the Flickr30k dataset in Appendix J.1 to demonstrate its cross-dataset generalization capability, our primary benchmarking is conducted on the standard transfer setting from the NIPS 2017 competition dataset to MS COCO to ensure fair comparison with baseline methods. Future work should investigate broader domain transfer to more comprehensively demonstrate its universal robustness. Furthermore, although we incorporated cross-judge validation to mitigate biases when using LLM-as-a-judge, we acknowledge that automated metrics may remain sensitive to specific prompts.

## J. Additional Experiments on Generalization and Robustness

To further assess the robustness and generalization of DART, we performed additional experiments beyond the standard NIPS 2017 competition dataset→MS COCO benchmark.

### J.1. Cross-Dataset Generalization (Flickr30k)

To assess transferability across different data distributions, we conducted transfer attacks on a new target dataset, Flickr30k (Young et al., 2014), using the same source images from the NIPS 2017 competition dataset. We randomly selected 200 images from Flickr30k, ensuring they did not overlap with previously used images, to form 200 source-target pairs, and evaluated the performance on GPT-4o and Gemini-2.0.

*Table 11.* Performance comparison on NIPS → Flickr30k.

| Method | Model | GPT-4o | | Gemini-2.0 | |
|---|---|---|---|---|---|
| | | ASR | AvgSim | ASR | AvgSim |
| FOA-Attack | Ensemble | 83.5 | 0.46 | 70.5 | 0.38 |
| Ours | Ensemble | $85.0_{\uparrow 1.5}$ | $0.48_{\uparrow 0.02}$ | $77.5_{\uparrow 7.0}$ | $0.41_{\uparrow 0.03}$ |

As shown in Table 11, DART consistently outperforms the baseline model across various target models under this configuration. The substantial improvement observed on Gemini-2.0 demonstrates that our approach captures general semantic patterns, which remain unaffected by domain variations.

### J.2. Sensitivity to Perturbation Budget ($\epsilon$)

We analyzed the sensitivity of the DART algorithm to the perturbation budget, $\epsilon$. As shown in Table 12, under stringent constraints ($\epsilon = 8/255$), the DART algorithm still outperforms the baseline, demonstrating that distribution alignment is more robust than point-wise matching in limited optimization spaces. Performance approaches saturation at $\epsilon = 32/255$, indicating that the DART algorithm is highly efficient under standard budgets.

*Table 12.* Sensitivity analysis of perturbation budget $\epsilon$ on GPT-4o.

| Budget ($\epsilon$) | Method | ASR (%) |
|---|---|---|
| 8/255 (Low) | FOA-Attack | 64.5 |
| | Ours | 68.0 |
| 16/255 (Standard) | Ours | 89.5 |
| 32/255 (High) | Ours | **90.0** |

