# OpenReview forum: "DART: Distribution-Aware Adaptive Relational Transfer for Adversarial Attacks against Closed-Source MLLMs"
_ICML.cc/2026/Conference — ICML 2026 regular_

### Official Review · Reviewer_kGWM · 2026-03-10

**Soundness:** 2
**Presentation:** 3
**Significance:** 4
**Originality:** 3
**Overall Recommendation:** 4
**Confidence:** 3

**Summary:**

This paper focuses on adversarial sample attacks against multimodal large language models (MLLMs). It generates perturbations by aligning latent distributions and utilizes Energy Distance to capture the image semantic manifold, thereby enhancing generalization ability. The proposed method can not only capture main objects but also identify fine-grained environmental details, endowing it with stronger robustness and transferability. Meanwhile, the authors train a GNN Attack Router, which adopts a data-driven approach to replace heuristic ensemble strategies for selecting dedicated surrogate models. This effectively identifies the optimal ensemble of task-specific attack surrogate models and improves the specialization of attacks. The paper includes sufficient experiments and appendix proofs, demonstrating high completeness. After the experiments, the authors also provide detailed explanations of the intrinsic causes, along with comprehensive ablation studies and analyses of robustness and effectiveness.

**Compliance With Llm Reviewing Policy:**

Affirmed.

**Key Questions For Authors:**

1.  Regarding the contradiction in Eq. (1): Please explicitly explain the physical meaning of maximizing the intra-distribution distance (i.e., maximizing $E[d_c(X, X')]$) when minimizing the Energy Distance in this specific adversarial attack task. Does this behavior risk causing semantic divergence or collapse in the generated adversarial samples?
2.  Regarding the performance at $M=1$: Please provide the exact ASR and AvgSim values for $M=1$ under the exact experimental settings of Figure 4. How significant is the performance jump specifically from $M=1$ to $M=2$?
3.  Regarding the true nature of "Distribution Alignment": Given that $M=5$ is statistically insufficient to fit a true distribution in a high-dimensional space, do you agree that the performance gain primarily originates from escaping sharp local minima via gradient smoothing (similar to EoT)? Furthermore, if $M$ is pushed to 10, 50, or 100, would the performance continue to rise, plateau, or stagnate due to the narrow semantic manifold of simple data augmentations? Please discuss this theoretical approximation vs. computational limitation trade-off.
4.  Regarding VRAM: Under the default configuration ($M=5, K=2$), what is the Peak VRAM required to generate a single adversarial sample? By what percentage does this increase compared to FOA-Attack?
This paper excels in engineering execution and empirical results, and the introduction of the GNN Attack Router is a valuable innovation. However, in explaining its core contribution ("Distribution-Aware Intrinsic Mining"), there is a fundamental conflict between the theoretical formula (minimization behavior of Energy Distance) and intuition. Furthermore, the experimental design for the core hyperparameter $M$ appears to evade extreme values (both the $M=1$ baseline and very large $M$), leading to a notable degree of "overclaiming" regarding distribution estimation. If the authors can satisfactorily resolve the intra-distance maximization issue in Equation 1, provide the $M=1$ ablation data, and appropriately recalibrate their theoretical claims regarding high-dimensional distribution estimation in the Rebuttal, I would be highly willing to raise my score.

**Limitations:**

Yes

**Strengths And Weaknesses:**

Technically, the paper is rigorous, and its research conclusions are well-supported by rigorous theoretical analyses and sufficient experimental results. The adopted research methods are appropriate and innovative. The paper is clearly written with a reasonable structure, making it easy to understand overall. The research also clarifies its own research value and differences from existing literature. Focusing on the key issue of MLLMs’ vulnerability, this research promotes the development of the field and exerts valuable impacts on future related research and applications. Nevertheless, there is still room for improvement in the presentation to enhance readability.

---

> ### Author Rebuttal · Authors · 2026-03-30
>
> We are truly grateful for your insightful review. Here we address your comments in the following.
>
> > Q1. Physical meaning of maximizing intra-distribution distance in Eq.(1).
>
> A1. Thank you for your insightful observation. We would like to clarify that this concern does not hold in practice.
>
> (1) For **global features**, we use the point-to-distribution (p→d) Energy Distance (Section 3.2). On the adversarial-sample side, the distribution’s internal term, $\mathbb{E}[d_c(X, X')]$, is zero because it corresponds to a single point (a Dirac delta distribution). Therefore, global alignment **does not involve maximizing intra-distribution distance**.
>
> (2) For **local features**, the intra-distribution terms $E_{uu}$ and $E_{vv}$ do participate in the computation. However, they serve as a **calibration mechanism** that prevents feature collapse, a property that makes the energy distance a strictly proper scoring rule (Appendix A, Theorem). If all adversarial local features collapse to a single point, $E_{uu} \to 0$ increases the overall Energy Distance, thereby steering the optimization away from such degenerate solutions.
>
> (3) **Empirical verification (on GPT-4o)**: We tracked $E_{uu}(L)$ across 300 optimization steps:
>
> | Step | $E_{cross}(G)$ | $E_{uu}(L)$ | $E_{vv}(L)$ | Loss |
> | ---- | -------------- | ----------- | ----------- | ---- |
> | 0    | 0.671          | 0.575       | 0.576       | 5.87 |
> | 100  | 0.515          | 0.577       | 0.594       | 3.81 |
> | 300  | 0.233          | 0.578       | 0.594       | 1.74 |
>
> $E_{uu}(L)$ remained stable at approximately 0.57 throughout the optimization process, while $E_{cross}(G)$ steadily decreased. The intra-distribution term neither diverged nor collapsed. Instead, it converged to a value matching the target’s internal variance (approximately 0.59), which is fully consistent with theoretical predictions.
>
> > Q2. Performance at M=1 and the jump from M=1 to M=2.
>
> A2. Thank you for your question. We provide the requested results on GPT-4o (M=2 from Table 5. M=1, M=20 are new):
>
> | M    | ASR   | AvgSim |
> | ---- | ----- | ------ |
> | 1    | 83.0% | 0.51   |
> | 2    | 86.0% | —      |
> | 5    | 89.5% | 0.53   |
> | 20   | 90.5% | 0.52   |
>
> The ASR improves by **3.0% from M=1 to M=2** and by **6.5% from M=1 to M=5**. When M=1, the Energy Distance degenerates into a point-to-point distance because the intra-distribution term vanishes. Even at M=2, significant gains are already observed, and performance tends to plateau after M=5, confirming that M=5 is the best trade-off point.
>
> > Q3. Is the performance gain primarily from gradient smoothing (EoT-like)?
>
> A3. Thank you for your question. We conducted controlled ablation studies on GPT-4o. We constructed an AvgCosine baseline that is identical to DART (same M = 5, GNN router, and dynamic weighting) but replaces the energy distance with the average cosine distance by removing the $E_{uu}$ and $E_{vv}$ terms. This baseline is mathematically equivalent to EoT.
>
> | Method                | Loss Function                      | ASR   | AvgSim |
> | --------------------- | ---------------------------------- | ----- | ------ |
> | DART (Ours)           | Energy Distance                    | 89.5% | 0.53   |
> | AvgCosine (EoT)       | Avg cosine (no $E_{uu}$, $E_{vv}$) | 82.5% | 0.45   |
> | M=1 (no augmentation) | Point-wise                         | 83.0% | 0.51   |
>
> The EoT baseline (82.5%) performs comparably to M=1 (83.0%), indicating that pure gradient smoothing yields little to no benefit. In contrast, DART outperforms EoT by 7.0% in ASR and 0.08 in AvgSim, clearly showing that the intra-distribution calibration term provides substantial value **beyond gradient smoothing**.
>
> We acknowledge that, with M=5 in a high-dimensional space, we approximate rather than perfectly recover the true distribution. In the final version, we will revise the wording to state “distribution-aware semantic alignment under finite-sample approximation” rather than implying exact distribution matching.
>
> > Q4. VRAM comparison.
>
> A4. Thank you for your question.
>
> | Configuration   | Peak VRAM | Time/image |
> | --------------- | --------- | ---------- |
> | DART (M=5, K=2) | 18.9 GB   | 175s       |
> | DART (M=2, K=2) | 18.9 GB   | 86s        |
> | FOA-Attack      | 7.8 GB    | 109s       |
>
> The VRAM increase is not caused by distribution estimation. The GPU memory requirements for M=2 and M=5 are identical. The additional overhead comes from loading the GNN router infrastructure, which is a one-time fixed cost. Crucially, in terms of computation time, DART (M=2) is **1.3 times faster** than FOA-Attack (86 s vs. 109 s, Table 5) while achieving a higher ASR (86.0% vs. 85.5%).
>
> In light of these responses, we hope we have addressed your concerns, and hope you will consider raising your score. If there are any additional notable points of concern that we have not yet addressed, please do not hesitate to share them, and we will promptly attend to those points.

---

> > ### Author Rebuttal · Reviewer_kGWM · 2026-04-04
> >
> > I would like to sincerely thank the authors for their thorough, professional, and highly convincing rebuttal. The response directly and perfectly addresses all of the core concerns I raised during the initial review. Based on the theoretical clarifications and the additional ablation studies provided, my evaluation of this paper has significantly improved. I strongly advise the authors to incorporate the core elements of this rebuttal into the final manuscript (either in the main text or the appendix). Specifically, the dynamic tracking of $E_{uu}$ (from A1), the expanded $M$ ablation trend (from A2), and the crucial `AvgCosine` baseline comparison (from A3) significantly elevate the theoretical depth and rigor of the paper. These additions will be immensely valuable to future readers in understanding the true mechanics of DART.
> >
> > Congratulations on an outstanding piece of work!

---

> > > ### Author Response · Authors · 2026-04-05
> > >
> > > We sincerely appreciate your thorough evaluation and highly encouraging feedback. Your constructive comments have been invaluable in improving our work. As you suggested, we will incorporate the key elements from our rebuttal, including the dynamic tracking of $E_{uu}$ (A1), the expanded $M$ ablation trend (A2), and the AvgCosine baseline comparison (A3), into the final manuscript. If you feel the clarified evidence and revisions strengthen the work overall, we would of course be grateful if you could consider reflecting that in your final assessment. Please let us know if you have any further concerns.

---

### Official Review · Reviewer_vSMv · 2026-03-11

**Soundness:** 3
**Presentation:** 4
**Significance:** 4
**Originality:** 3
**Overall Recommendation:** 4
**Confidence:** 4

**Summary:**

This paper proposes a novel framework DART to improve the generalization ability and adaptability of attacks on closed source MLLMs. This framework decomposes the adversarial attack problem into generalization and specialization: using energy distance for distribution level alignment to alleviate overfitting of surrogate models, utilizing graph neural network-based (GNN) attack routers for offline learning and adaptive selection. Experiments on both open-source and closed-source MLLMs demonstrate the effectiveness of the proposed method.

**Compliance With Llm Reviewing Policy:**

Affirmed.

**Final Justification:**

I sincerely thank the authors for their rebuttal. Nonetheless, my original rating remains unchanged.

**Key Questions For Authors:**

See weaknesses.

**Limitations:**

Yes.

**Strengths And Weaknesses:**

Strengths:
1. The topic is innovative and the research has practical significance.
2. Novel distribution-level alignment with rigorous theoretical analysis.
3. This paper systematically evaluates each component and conducts a clear trade-off analysis between performance and computational cost.
4. The figure and writing are clear and the paper is easy to follow.
Weaknesses:
1. This paper focuses entirely on attack performance and lacks evaluation against adversarial defenses.

---

> ### Author Rebuttal · Authors · 2026-03-30
>
> We are truly grateful for the time you have taken to review our paper, your insightful comments and support. Your positive feedback is incredibly encouraging for us! In the following response, we would like to address your major concern and provide additional clarification.
>
> > W1. Lack of evaluation against adversarial defenses.
>
> A1. Thank you for this important suggestion. We have conducted a comprehensive evaluation of defensive measures. Before feeding adversarial examples into GPT-4o, we applied three common input preprocessing defenses and compared DART with FOA-Attack.
>
> | Defense | DART ASR | FOA ASR | DART Advantage |
> |---------|----------|---------|----------------|
> | No defense | 89.5% | 85.5% | +4.0% |
> | JPEG compression (Q=75) | 87.5% | 80.0% | +7.5% |
> | Gaussian Blur (r=1.0) | 65.0% | 53.0% | +12.0% |
> | Bit-depth reduction (4-bit) | 84.0% | 82.5% | +1.5% |
>
> Key findings:
>
> (1) **DART consistently outperforms FOA-Attack under all defense settings.** This suggests that distribution-aligned perturbations are inherently more robust than point-wise matching perturbations.
>
> (2) **The advantage widens under stronger defenses.** Under Gaussian blur, which is the most aggressive defense, DART’s lead expanded from +4.0% under no defense to +12.0%. This is because distribution-aligned perturbations capture more robust semantic features that better withstand input transformations, whereas pixel-wise perturbations are more vulnerable to such distortions.
>
> (3) **Practical robustness.** Under JPEG-75, a defense commonly used in real-world systems, DART maintains an ASR of 87.5%, a decline of only 2.0% compared with the no-defense setting, demonstrating strong robustness in practical settings.
>
> We also note that closed-source commercial models likely incorporate built-in defense mechanisms. Our strong ASR results on these models (Tables 1–3) implicitly indicate robustness against these built-in defenses.
>
> We will incorporate these defense evaluation results into the final version. We sincerely appreciate your suggestion, as it strengthens the paper’s contributions.
>
> Thanks again for appreciating our work and for your constructive suggestions. Please let us know if you have further questions.

---

> > ### Author Rebuttal · Reviewer_vSMv · 2026-04-04
> >
> > Thank you for the response. My major concerns have been addressed. I keep my original rating.

---

> > > ### Author Response · Authors · 2026-04-05
> > >
> > > We sincerely appreciate your time in carefully reviewing our rebuttal and confirming that your major concerns have been addressed. We are glad the defense evaluation results helped strengthen the paper. Thank you again for your valuable feedback and continued support.

---

### Official Review · Reviewer_s874 · 2026-03-12

**Soundness:** 4
**Presentation:** 4
**Significance:** 3
**Originality:** 3
**Overall Recommendation:** 4
**Confidence:** 5

**Summary:**

This paper tackles the challenging problem of targeted transfer attacks against black-box Multimodal Large Language Model. The authors propose DART, a novel framework that generates an imperceptible perturbation to force a target MLLM to describe an image as a completely different, attacker-chosen target. DART introduces two key innovations to overcome the limitations of prior point-wise feature matching: Distribution-aware Intrinsic Mining , which aligns the distributions of augmented views of images using Energy Distance to improve generalization, and a GNN-based Attack Router that learns to select the most effective combination of surrogate models from historical data. Extensive experiments on a wide range of open-source and closed-source MLLMs demonstrate that DART consistently and significantly outperforms existing state-of-the-art attack methods.

**Compliance With Llm Reviewing Policy:**

Affirmed.

**Final Justification:**

My concerns have been adequately addressed

**Key Questions For Authors:**

1. The GNN router is trained on logs from attacks against a specific target (Qwen2-VL-7B). How sensitive is its performance to the choice of this log-collection target? Would performance degrade if the router were trained on logs from a weaker model and then used to select surrogates for attacking a much stronger one (or vice-versa)?

2.The choice of data augmentations seems crucial for defining a meaningful "semantic distribution." How were these specific augmentations chosen? Is there a risk that for certain target images, this set of augmentations fails to capture the true semantic manifold, leading to a poor distribution estimate? Similarly, what clustering algorithm  was used for local features, and what was the rationale behind this choice?

**Limitations:**

1.As noted in the weaknesses, the router's effectiveness is constrained by its offline training data. Adapting to new, unseen model architectures without retraining remains an open challenge.

2.The distribution estimation via multiple augmentations introduces non-trivial computational costs. While a faster variant is offered, the best-performing model is computationally intensive, potentially limiting its use in very high-throughput or real-time attack scenarios.

3.The threat model is confined to the digital space with  norm constraints. The paper does not address the more challenging physical-world attack scenario where perturbations must survive printing, re-capturing, and other real-world distortions. The robustness of the distribution alignment concept in such conditions is an important avenue for future work.

4.The attack focuses on semantic alignment of the entire image description. It does not demonstrate finer-grained control, such as forcing the model to describe a specific, subtle attribute of the target image while ignoring others, or achieving a specific verbatim output. The scope is "what" is described, not necessarily "how" it is described in precise detail.

5.The method's reliance on CLIP-based encoders, while standard, means its effectiveness against MLLMs built on fundamentally different visual encoders or processing pipelines is not fully explored. The strong results on closed-source models are promising, but a direct investigation into the transferability from CLIP to, for instance, a model using a pure diffusion-based vision encoder would be a valuable stress test.

**Strengths And Weaknesses:**

Strengths:

1. The core idea of shifting the attack objective from point-wise feature matching to distribution-level alignment is both novel and theoretically sound. The paper clearly articulates the overfitting limitations of prior work and provides a compelling justification for why aligning distributions should lead to better transferability.

2.The experimental evaluation is extensive and rigorous. The authors evaluate against a wide array of both open-source.

3.The paper goes beyond simple metric reporting. Thorough ablation studies dissect the contribution of each component.

4.The paper is easy to follow.


Weaknesses:

1.The GNN Attack Router's effectiveness is entirely dependent on the quality and scale of the historical attack logs used for pre-training. The paper mentions using Qwen2-VL-7B as the "log-collection target," but details on the diversity of these logs are sparse. I think a lack of diverse historical data could lead to a router that overfits to specific transfer patterns and performs poorly on truly novel tasks or model architectures not represented in the training data.

2. While Table 5 shows a faster configuration, the full DART framework is complex. The computational overhead of the offline training and the online augmentation could be a barrier for researchers with limited resources, especially when compared to simpler ensemble methods.

3. Despite aiming for a data-driven approach, the framework still incorporates heuristic choices. The specific set of data augmentations used for distribution estimation is a critical design decision that is not learned. Similarly, the clustering algorithm for local features and the hyperparameter  balancing global/local losses, while ablated, are fixed per model and not dynamically adapted. The paper acknowledges this as a limitation (Section I.2), but I think it slightly tempers the claim of a fully "data-driven" system.

---

> ### Author Rebuttal · Authors · 2026-03-30
>
> We are truly grateful for the time you have taken to review our paper, your insightful comments and support. Your positive feedback is incredibly encouraging for us! In the following response, we would like to address your major concern and provide additional clarification.
>
> > Q1. Sensitivity of GNN Router to the log-collection target choice.
>
> A1. Thank you for your question. The GNN router learns *relative transfer patterns* among surrogate models, that is, which combinations of surrogate models perform best on which source-target image pairs, rather than features specific to any particular target MLLM. As described in Section 3.3, the training logs were collected using Qwen2-VL-7B, and all evaluation target models (GPT-4o, Gemini-2.0, etc.) are completely distinct from it. The strong out-of-sample results on these unseen models (Tables 1–3) empirically validate that the learned transfer relationships generalize across target architectures.
>
> We agree that investigating the sensitivity of our method to different log-collection target models is a valuable direction. We plan to explore this issue in future work, including training on logs from multiple target models to further improve robustness.
>
> > Q2. How were data augmentations chosen? What clustering algorithm was used?
>
> A2. Thank you for your question. The augmentations (RandomResizedCrop with scale=(0.5,1.0), RandomHorizontalFlip, ColorJitter) are standard semantic-preserving transformations validated in self-supervised learning. The design principle is that transformations should alter low-level visual features while preserving high-level semantics. This aligns with our assumption of “sampling from the intrinsic semantic distribution.” As discussed in Appendix I.2, automated augmentation selection is an important future direction.
>
> For the clustering algorithm, we use K-means with 10 clusters for local feature aggregation. K-means was chosen for its simplicity, guaranteed convergence, and computational efficiency in iterative training loops. The number of clusters (10) was selected to balance the granularity of local pattern representation and computational overhead.
>
> Additionally, we conducted a new ablation study on GPT-4o to examine whether augmentations provide benefits beyond gradient smoothing. We compared DART (Energy Distance, M=5) against an AvgCosine baseline with the same augmentations and M=5, but using average cosine distance, which is equivalent to EoT. The results show that DART achieves 89.5% ASR, compared with 82.5% for AvgCosine, confirming that the distributional formulation, not merely the augmentation choice, drives the performance gain.
>
> > W2. Computational overhead.
>
> Thank you for raising this concern. We provide additional VRAM and timing data:
>
> | Configuration | Peak VRAM | Time/image |
> |---------------|-----------|------------|
> | DART (M=5, K=2) | 18.9 GB | 175s |
> | DART (M=2, K=2) | 18.9 GB | 86s |
> | FOA-Attack | 7.8 GB | 109s |
>
> The increase in VRAM is due to loading the GNN router infrastructure, which introduces a one-time fixed overhead, rather than to distribution estimation. The VRAM requirements for M=2 and M=5 are identical. In terms of computation time, DART (M=2) is **1.3 times faster** than FOA (86 s vs. 109 s, Table 5) while achieving a higher ASR (86.0% vs. 85.5%).
>
> **Addressing Limitations L1–L5.**
>
> We appreciate these thoughtful observations. L1 (router data dependency) and L2 (computational overhead) have already been addressed in Q1 and W2 above. L4 (fine-grained control) has been discussed in Appendix I. Below, we present new results on L3 and L5.
>
> **L3 (robustness to real-world distortions):** We conducted a new adversarial defense evaluation on GPT-4o with three common input preprocessing defenses:
>
> | Defense | DART ASR | FOA ASR | DART Advantage |
> |---------|----------|---------|----------------|
> | JPEG-75 | 87.5% | 80.0% | +7.5% |
> | Gaussian Blur | 65.0% | 53.0% | +12.0% |
> | BitDepth-4 | 84.0% | 82.5% | +1.5% |
>
> DART consistently outperforms FOA under all defense settings, with a more pronounced advantage under stronger defenses (blurring: +12%). This suggests that distribution-aligned perturbations capture more robust semantic features that better withstand input transformations.
>
> Regarding **Limitation 5 (non-CLIP encoders)**: We agree that evaluating transferability to fundamentally different visual architectures, such as diffusion-based encoders, is a valuable stress test. While our pool of surrogate models aligns with prior work to ensure fair benchmarking, we recognize this as a critical next step that will be explicitly addressed in the final version.
>
> Thanks again for appreciating our work and for your constructive suggestions. Please let us know if you have further questions.

---

> > ### Author Rebuttal · Reviewer_s874 · 2026-04-03
> >
> > My concerns have been adequately addressed

---

> > > ### Author Response · Authors · 2026-04-03
> > >
> > > We sincerely appreciate your time in carefully reviewing our rebuttal and confirming that your concerns have been fully addressed. We greatly appreciate your constructive feedback, which will help improve our paper. Thank you again for your time and effort.

---

### Official Review · Reviewer_jufe · 2026-03-24

**Soundness:** 3
**Presentation:** 2
**Significance:** 2
**Originality:** 2
**Overall Recommendation:** 3
**Confidence:** 4

**Summary:**

This paper presents a new adversarial attack method on VLM, including two contributions: optimizing from the distribution perspective and adaptive relational mining for surrogate model selection. The experiments validate its effectiveness on several visual language models.

**Compliance With Llm Reviewing Policy:**

Affirmed.

**Key Questions For Authors:**

Q1: The novelty of this paper is limited, which can be treated as an extended version of FOA. This paper extends the latent measurement with additional distribution loss and ensemble methods with a GNN-based router.

Q2: What are the training details of the GNN-based router? What are the training parameters? What is the training data? It seems that you train the model on 80% of the testing data. Will this trigger overfitting to the test set?

Q3: What is the selected distribution of the ensemble models? In the appendix, you only use three ensemble models, and they tend to have similar structures (ViT). Why not select other vision encoders like Siglip? Is the router really useful? This is not validated because the ensembles are all from the same model structure family.

Q4: What is the transformation you considered? What are the parameters for those transformations?

**Limitations:**

yes

**Strengths And Weaknesses:**

Strength:

The topic is important.

The performance is good.

Weakness:

The details are not clear.

The novelty is limited.

---

> ### Author Rebuttal · Authors · 2026-03-30
>
> We are truly grateful for the time you have taken to review our paper and your insightful review. Here we address your comments in the following.
>
> > Q1. Novelty is limited, can be treated as an extended version of FOA.
>
> A1. Thank you for your comment. DART introduces three fundamental advances that go beyond FOA-Attack:
>
> - **Different optimization objective.** FOA-Attack aligns individual feature vectors via point-to-point optimal transport. DART models image semantics as a probability distribution and uses Energy Distance, a strictly proper scoring rule with mathematical guarantees ($D_E^2 = 0$ if and only if $P = Q$, see Appendix A). To validate this, we replaced Energy Distance with average cosine distance (equivalent to EoT-style gradient smoothing) while keeping all else identical: DART achieved 89.5% ASR vs. 82.5% for the average cosine baseline on GPT-4o (+7.0%), confirming the distributional formulation provides substantial value beyond simple feature matching.
>
> - **Different model selection mechanism.** FOA-Attack employs heuristic weighting based on loss convergence. DART introduces a GNN-based attack router that explicitly models task-dependent transfer relationships through graph-based relational learning. Our ablation (Figure 3) shows GNN routing significantly outperforms random selection and static ensemble methods, representing a fundamentally different paradigm for surrogate model selection.
>
> - **Different generalization capability.** On the most robust targets: Claude-3.7, DART (32.0%) vs. FOA (23.5%), **+36% relative improvement**. On Claude-Sonnet-4, DART (14.5%) vs. FOA (9.5%), **+53% relative improvement**. Such gains on the hardest models are inconsistent with a mere incremental extension.
>
>
> > Q2. Training details of GNN router? Training on 80% of testing data—overfitting?
>
> A2. Thank you for your question. We clarify that this concern stems from a misunderstanding.
>
> (1) **No data leakage.** As described in Section 4.1, we train the GNN router on the first 800 image pairs and test it on the remaining 200 pairs. The training and test sets are **completely disjoint**. This is a standard train/test split, not training on test data.
>
> (2) **Zero-shot generalization to unseen targets.** More importantly, the GNN router's training logs are collected using Qwen2-VL-7B as the log-collection target (Section 3.3), while the testing targets (GPT-4o, Gemini-2.0, etc.) are **entirely disjoint models** that the router has never seen. The strong results on these unseen closed-source models demonstrate genuine zero-shot generalization rather than overfitting.
>
> (3) **Training details** (Appendix B&D): Architecture: 2-layer HeteroConv (GraphSAGE) + MLP edge predictor, hidden_dim=128. Optimizer: Adam, lr=0.001. Loss: MSE. Early stopping: patience=20 on validation loss. Training data: 800 pairs × 3 surrogate models = 2400 attack log entries.
>
> > Q3. Only ViT-based surrogates. Why not SigLIP? Is the router really useful?
>
> A3. Thank you for your question.
>
> (1) **Fair comparison.** The pool (ViT-B/16, ViT-B/32, Laion-g/14) was chosen to align with prior work (M-Attack and FOA-Attack) for a **strictly fair comparison** (Appendix D).
>
> (2) **Architectural diversity within the pool.** Although these models share the ViT name, they differ substantially: ViT-B/16 and ViT-B/32 have ~86M visual parameters, whereas ViT-g/14-Laion has ~1B parameters and was trained on LAION-2B—a **nearly 12-fold scale difference** with distinct training distributions that the router exploits.
>
> (3) **Router utility is directly quantified in Figure 3.**
>
> | K | Strategy | GPT-4o ASR | Gemini-2.0 ASR |
> |---|----------|-----------|----------------|
> | 1 | Single model | 86.0% | 76.5% |
> | 3 | Static ensemble | 87.5% | 79.0% |
> | 2 | GNN Router (Ours) | **89.5%** | **80.5%** |
>
> This consistent advantage on the *same* ViT-family pool shows the router's value derives from task-adaptive learning, not architectural diversity. Extending to encoder families such as SigLIP is a valuable future direction.
>
> > Q4. What transformations and parameters?
>
> A4. Thank you for your question. The data augmentations for distribution estimation are: RandomResizedCrop(224, scale=(0.5,1.0)), RandomHorizontalFlip(p=0.5), and ColorJitter(brightness=0.4, contrast=0.4, saturation=0.4, hue=0.1, p=0.7), following standard semantic-preserving augmentation practice in self-supervised learning.
>
> **Regarding the "details not clear" concern:** We acknowledge this feedback. Key details (GNN parameters, augmentation specs, clustering, metrics) are in Appendices B, D, and E. We will consolidate them into the main text in the final version for better accessibility.
>
> In light of these responses, we hope we have addressed your concerns, and hope you will consider raising your score. If there are any additional notable points of concern that we have not yet addressed, please do not hesitate to share them, and we will promptly attend to those points.

---

> > ### Author Rebuttal · Reviewer_jufe · 2026-04-05
> >
> > After reading the authors' response, I decide to maintain the original score.

---

> > > ### Author Response · Authors · 2026-04-06
> > >
> > > We thank the reviewer for reading our response and for your time throughout the review process. If you have any further concerns or follow-up questions, please let us know. We really appreciate your feedback.

---

### Decision · Program_Chairs · 2026-04-30

**Decision:**

Accept (regular)

**Comment:**

The paper introduces DART, a novel framework for adversarial attacks on MLLMs, combining distribution-level alignment with an adaptive GNN-based attack router. Reviewers broadly agree that the approach is technically sound, well-motivated, and demonstrates strong empirical performance across both open- and closed-source models. The shift from point-wise matching to distribution-aware optimization is viewed as a meaningful and effective contribution that improves transferability. Extensive experiments, ablations, and analyses further support the validity and general applicability of the proposed method. While concerns remain regarding clarity of some details, computational overhead, and reliance on offline data for the router, these are not deemed critical to the core contribution. Overall, given the solid technical quality, consistent gains over prior work, and relevance of the problem, the paper shall be accepted.